# Improved Off-policy Reinforcement Learning in Biological Sequence Design

**Hyeonah Kim** [1 2]   **Minsu Kim** [1 3]   **Taeyoung Yun** [3]   **Sanghyeok Choi** [3]   **Emmanuel Bengio** [4]
**Alex Hernández-García** [1 2]   **Jinkyoo Park** [3]

## Abstract

Designing biological sequences with desired properties is challenging due to vast search spaces and limited evaluation budgets. Although reinforcement learning methods use proxy models for rapid reward evaluation, insufficient training data can cause proxy misspecification on out-of-distribution inputs. To address this, we propose a novel off-policy search, $\delta$-Conservative Search, that enhances robustness by restricting policy exploration to reliable regions. Starting from high-score offline sequences, we inject noise by randomly masking tokens with probability $\delta$, then denoise them using our policy. We further adapt $\delta$ based on proxy uncertainty on each data point, aligning the level of conservativeness with model confidence. Experimental results show that our conservative search consistently enhances the off-policy training, outperforming existing machine learning methods in discovering high-score sequences across diverse tasks, including DNA, RNA, protein, and peptide design.

## 1. Introduction

Designing biological sequences with desired properties is crucial in therapeutics and biotechnology (Zimmer, 2002; Lorenz et al., 2011; Barrera et al., 2016; Sample et al., 2019; Ogden et al., 2019). However, this task is challenging due to the combinatorially large search space and the expensive and black-box nature of objective functions. Recent advances in deep learning methods for biological sequence design have shown significant promise at overcoming these challenges (Brookes & Listgarten, 2018; Brookes et al., 2019; Angermueller et al., 2020; Jain et al., 2022).

Among various approaches, reinforcement learning (RL)

---
[1]Mila - Quebec AI Institute [2]Université de Montréal [3]KAIST [4]Valence Labs. Correspondence to: Hyeonah Kim <hyeonah.kim@mila.quebec>.

*Proceedings of the 42$^{nd}$ International Conference on Machine Learning*, Vancouver, Canada. PMLR 267, 2025. Copyright 2025 by the author(s).

has emerged as one of the successful paradigms for automatic biological sequence design (Angermueller et al., 2020; Jain et al., 2022). Particularly, they employ deep neural networks as inexpensive proxy models for rapid reward evaluation by approximating expensive oracle score functions. There are two approaches for training the policy in this context: on-policy and off-policy.

DyNA PPO (Angermueller et al., 2020), a representative on-policy RL method for biological sequence design, employs Proximal Policy Optimization (PPO; Schulman et al., 2017) within a proxy-based active training loop. While DyNA PPO has demonstrated effectiveness in various biological sequence design tasks, its major limitation is the limited search flexibility inherent to on-policy methods. It cannot effectively leverage offline data points, even like data collected from previous rounds.

On the other hand, Generative Flow Networks (GFlowNets; Bengio et al., 2021), off-policy RL methods akin to maximum entropy policies (Tiapkin et al., 2024; Deleu et al., 2024), offer diversity-seeking capabilities and flexible exploration strategies. Jain et al. (2022) applied GFlowNets to biological sequence design with additional Bayesian active learning schemes. They leveraged the off-policy nature of GFlowNets by mixing offline datasets with on-policy data during training. This approach provided more stable training compared to DyNA PPO, resulting in better performance.

However, recent studies have consistently reported that GFlowNets perform poorly in large-scale settings, such as green fluorescent protein design (Kim et al., 2023; Surana et al., 2024). We hypothesize that this bounded performance stems from the insufficient quality of the proxy model in the early rounds. Although GFlowNets can generate novel sequences beyond given data points, the resulting out-of-distribution inputs lead to unreliable rewards from the undertrained proxy (Trabucco et al., 2021; Yu et al., 2021). This motivates us to introduce a conservative search strategy to restrict the search space to the neighborhoods of the data points observed during RL training when generating sequences to query for the next round.

**Contribution.** In this paper, we propose a novel off-policy search method called $\delta$-Conservative Search ($\delta$-CS), which enables a trade-off between sequence novelty and robustness

to proxy misspecification with a conservativeness parameter $\delta$. Specifically, we iteratively train a GFlowNet using $\delta$-CS as follows: (1) we inject noise by independently masking tokens in high-score offline sequences with a Bernoulli distribution with parameter $\delta$; (2) the GFlowNet policy sequentially denoises the masked tokens; (3) we use these denoised sequences to train the policy. Figure 1 illustrates the overall procedure of $\delta$-CS. We adjust the conservative level with adaptive $\delta(x; \sigma)$ using the proxy model's uncertainty estimates $\sigma(x)$ for each data point $x$.

Our extensive experiments demonstrate that $\delta$-CS significantly improves GFlowNets, successfully discovering higher-score sequences compared to existing model-based optimization methods on diverse tasks, including DNA, RNA, protein, and peptide design. This suggests that $\delta$-CS is a robust and scalable framework for advancing research and applications in biotechnology and synthetic biology.

## 2. Problem Formulation

We aim to discover sequences $x \in \mathcal{V}^L$ that exhibit desired properties, where $\mathcal{V}$ denotes the vocabulary, such as amino acids or nucleotides, and $L$ represents the sequence length, which is usually fixed. The desired properties, such as binding affinity or enzymatic activity, are evaluated by a black-box oracle function $f : \mathcal{V}^L \to \mathbb{R}$. Evaluating $f$ is often both expensive and time-consuming since it typically involves wet-lab experiments or high-fidelity simulations.

Advancements in experimental techniques have enabled the parallel synthesis and evaluation of sequences in batches. Therefore, lab-in-the-loop processes are emerging as practical settings that enable active learning. Following this paradigm, we perform $T$ rounds of batch optimization, where in each round, we have the opportunity to query $B$ batched sequences to the (*assumed*) oracle objective function $f$. Following Angermueller et al. (2020) and Jain et al. (2022), we assume the availability of an initial offline dataset $\mathcal{D}_0 = \{(x^{(n)}, y^{(n)})\}_{n=1}^{N_0}$, where $y = f(x)$. The initial number of data points $N_0$ is typically many orders of magnitude smaller than the size of the search space, as mentioned in the introduction. The goal is to discover, after $T$ rounds, a set of sequences that are novel, diverse, and have high oracle function values.

## 3. Active Learning for Biological Sequence Design

Following Jain et al. (2022), we formulate an active learning process constrained by a budget of $T$ rounds with query size $B$. The active learning is conducted through an iterative procedure consisting of three standard stages, two of which are modified by our proposed method, $\delta$-Conservative Search ($\delta$-CS), which will be detailed in Section 4.

**Step A (Proxy Training):** We train a proxy model $f_\phi(x)$ using the offline dataset $\mathcal{D}_{t-1}$ at round $t$.

**Step B (Policy Training with $\delta$-CS):** We train a generative policy $p(x; \theta)$ using the proxy model $f_\phi(x)$ and the dataset $\mathcal{D}_{t-1}$ with $\delta$-CS.

**Step C (Offline Dataset Augmentation with $\delta$-CS):** We apply $\delta$-CS to query batched data $\{x_i\}_{i=1}^B$ to the oracle $y_i = f(x_i)$. Then the offline dataset is augmented as: $\mathcal{D}_t \leftarrow \mathcal{D}_{t-1} \cup \{(x_i, y_i)\}_{i=1}^B$.

The overall algorithm is described in Algorithm 1. In the following subsections, we describe the details of **Step A** and **Step B**.

### 3.1. Step A: Proxy Training

Following Jain et al. (2022), we train the proxy model $f_\phi$ using the dataset $\mathcal{D}_{t-1}$ by minimizing the mean squared error loss:

$$\mathcal{L}(\phi) = \mathbb{E}_{x \sim P_{\mathcal{D}_{t-1}}(x)} \left[ (f(x) - f_\phi(x))^2 \right], \quad (1)$$

where $\mathcal{D}_t$ is the dataset at active round $t$, augmented with oracle queries. In the initial round ($t = 1$), we use the given initial dataset $\mathcal{D}_0$. See Appendix A.1 for detailed implementation.

### 3.2. Step B: Policy Training with $\delta$-CS

For policy training, we employ GFlowNets,[1] which aim to produce samples from a generative policy where the probability of generating a sequence $x$ is proportional to its reward, i.e.,

$$p(x; \theta) \propto R(x; \phi) = \mathcal{F}(x; \phi)$$

Here, $\mathcal{F}(x; \phi)$ is an acquisition function. The proxy $f_\phi(x)$ can be directly used as rewards, but we mainly employ the upper confidence bound (UCB; Srinivas et al., 2010) acquisition function following Jain et al. (2022).

**Policy parameterization.** The forward policy $P_F$ generates state transitions sequentially within trajectories $\tau = (s_0 \to \ldots \to s_L = x)$, where $s_0 = ()$ represents the empty sequence, and each state transition involves adding a sequence token. The full sequence $s_L = x$ is obtained after $L$ steps, where $L$ is the sequence length. The forward policy $P_F(\tau; \theta)$ is a compositional policy defined as

$$P_F(\tau; \theta) = \prod_{i=1}^{L} P_F(s_i | s_{i-1}; \theta). \quad (2)$$

---

[1]Our focus is primarily on the use of GFlowNets as an off-policy reinforcement learning framework; see Section 5.2 and Appendix B.2 for further discussion.

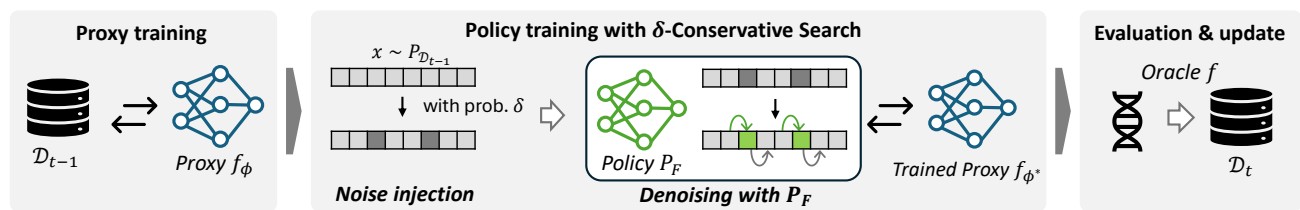

*Figure 1.* The active learning process for biological sequence design with $\delta$-Conservative Search ($\delta$-CS). Starting with high reward sequences from the offline dataset, we inject token-level noise with probability $\delta$, which determines the conservativeness of the search. Then, the GFlowNet policy denoises the masked sequences. Lastly, the GFlowNet policy is trained with new sequences. After policy training, we query a new batch of sequences and update the dataset for the next round.

GFlowNets have a backward policy $P_B(\tau|x)$ that models the probability of backtracking from the terminal state $x$. The sequence $x = (e_1, \ldots, e_L)$ can be uniquely converted into a state transition trajectory $\tau$, where each intermediate state represents a subsequence. In the case of sequences, there is only a single way to backtrack, so $P_B(\tau|x) = 1$. This makes these types of GFlowNets equivalent to soft off-policy RL algorithms. For example, the trajectory balance (TB) objective of GFlowNets (Malkin et al., 2022) becomes equivalent to path consistency learning (PCL) (Nachum et al., 2017), an entropy-maximizing value-based RL method according to Deleu et al. (2024).

**Learning objective and training trajectories.** The policy is trained to minimize TB loss as follows.

$$\mathcal{L}_{\text{TB}}(\tau; \theta) = \left( \log \frac{Z_\theta P_F(\tau; \theta)}{R(x; \phi)} \right)^2 \quad (3)$$

Usually, GFlowNets training is employed to minimize TB loss with training trajectories $\tau$ on full supports, asymptotically guaranteeing optimality for the distribution of $p(x; \theta) \propto R(x; \phi)$. A key challenge in prior works Jain et al. (2022) is that the proxy model $f_\phi(x)$ often produces highly unreliable rewards $R(x; \phi)$ for out-of-distribution inputs. In our approach, we mitigate this by providing off-policy trajectories within more reliable regions by injecting conservativeness into off-policy search. Therefore, **we minimize TB loss with $\delta$-CS**, which offers controllable conservativeness.

## 4. Controllable Conservativeness in Off-Policy Search

### 4.1. $\delta$-Conservative Search

This section introduces $\delta$-Conservative Search ($\delta$-CS), an off-policy search method that enables controllable exploration through a conservative parameter $\delta$. Here, $\delta$ defines the Bernoulli distribution governing the masking of tokens in a sequence. Our algorithm proceeds as follows:

- Sample high-score offline sequences $x \sim P_{\mathcal{D}_{t-1}}(x)$ from

the **rank-based reweighted prior**.

- Inject noise by masking tokens into $x$ using the **noise injection policy** $P_{\text{noise}}(\tilde{x} \mid x, \delta)$.

- Denoise the masked tokens using the **denoising policy** $P_{\text{denoise}}(\hat{x} \mid \tilde{x}; \theta)$.

These trajectories are used to update the GFlowNet parameters $\theta$ by minimizing the loss function $\mathcal{L}_{\text{TB}}(\tau; \theta)$. For more details on the algorithmic components of $\delta$-CS and its integration with active learning GFlowNets, see Algorithm 1.

**Rank-based reweighted prior.** First, we sample a reference sequence $x$ from the prior distribution $P_{\mathcal{D}_{t-1}}$. To exploit high-scoring sequences, we employ rank-based prioritization (Tripp et al., 2020):

$$w(x; \mathcal{D}_{t-1}, k) \propto \frac{1}{kN + \text{rank}_{f, \mathcal{D}_{t-1}}(x)}.$$

Here, $\text{rank}_{f, \mathcal{D}_{t-1}}(x)$ is a relative rank of the value of $f(x)$ in the dataset $\mathcal{D}_{t-1}$ with a weight-shifting factor $k$; we fix $k = 0.01$. This assigns greater weight to sequences with higher ranks. Note that this reweighted prior can also be used during proxy training.

**Noise injection policy.** Let $x = (e_1, e_2, \ldots, e_L)$ denote the original sequence of length $L$. We define a noise injection policy where each position $i \in \{1, 2, \ldots, L\}$ is independently masked according to a Bernoulli distribution with parameter $\delta \in [0, 1]$, resulting in the masked sequence $\tilde{x} = (\tilde{e}_1, \tilde{e}_2, \ldots, \tilde{e}_L)$. The noise injection policy $P_{\text{noise}}(\tilde{x} \mid x, \delta)$ is defined as:

$$P_{\text{noise}}(\tilde{x} \mid x, \delta)$$
$$= \prod_{i=1}^{L} [\delta \cdot \mathbb{I}\{\tilde{e}_i = [\text{MASK}]\} + (1 - \delta) \cdot \mathbb{I}\{\tilde{e}_i = e_i\}],$$

where $\mathbb{I}\{\cdot\}$ is the indicator function.

**Denoising policy.** We employ the GFlowNet forward policy $P_F$ to sequentially reconstruct the masked sequence

$\tilde{x} = (\tilde{e}_1, \tilde{e}_2, \ldots, \tilde{e}_L)$ by predicting tokens from left to right. The probability of denoising next token $\tilde{e}_t$ from previously denoised subsequence $\hat{s}_{t-1}$ is:

$$P_{\text{denoise}}(\hat{e}_t \mid \hat{s}_{t-1}, \tilde{x}; \theta)$$
$$= \begin{cases} \mathbb{I}\{\hat{e}_t = \tilde{e}_t\}, & \text{if } \tilde{e}_t \neq [\text{MASK}], \\ P_F(\hat{s}_t = (\hat{s}_{t-1}, \hat{e}_t) \mid \hat{s}_{t-1}; \theta), & \text{if } \tilde{e}_t = [\text{MASK}]. \end{cases}$$

The fully reconstructed sequence $\hat{x} = \hat{s}_L$ is obtained by sampling from:

$$P_{\text{denoise}}(\hat{x} \mid \tilde{x}; \theta) = \prod_{t=1}^{L} P_{\text{denoise}}(\hat{e}_t \mid \hat{s}_{t-1}, \tilde{x}; \theta).$$

By denoising the masked tokens with the GFlowNet policy, which infers each token sequentially from left to right, we generate new sequences $\hat{x}$ that balance novelty and conservativeness through the parameter $\delta$.

### 4.2. Adjusting Conservativeness

Determining the conservativeness parameter $\delta$ is crucial. This work proposes a simple but effective way to adjust $\delta$ based on the uncertainty of the proxy on each sequence $x$. The key insight is that even though the proxy predicts the score inaccurately, its uncertainty, which measures how the proxy gives predictions inconsistently, can guide how conservatively we search. Specifically, we define a function that assigns lower $\delta$ values for highly uncertain samples and vice versa: $\delta(x; \sigma) = \delta_{\text{const.}} - \lambda\sigma(x)$, where $\lambda$ is a scaling factor. We clamp the result to ensure it fall in $[0, 1]$.

In this adaptive $\delta$ control, the reduced $\delta$ leads to a more conservative search if the proxy gives inconsistent predictions. We estimate $\sigma(x)$, the standard deviation of the proxy model $f_\phi(x)$, via MC dropout (Gal & Ghahramani, 2016) or an ensemble method (Lakshminarayanan et al., 2017). Note that we measure the uncertainty on the observed data points. In practice, $\delta_{\text{const.}}$ is set as 0.5 for short sequences and 0.05 for longer ones, such as proteins, as $\delta L$ tokens are masked on average. While more tailored choices of $\delta_{\text{const.}}$ could be made with task-specific knowledge, our empirical findings indicate that these simple defaults work well across a variety of settings; see Section 6.

## 5. Related Work

### 5.1. Biological Sequence Design

Designing biological sequences using machine learning methods is widely studied. Bayesian optimization (BO) methods (Mockus, 2005; Belanger et al., 2019; Zhang et al., 2022) exploit posterior inference over newly acquired data points to update a Bayesian proxy model that can measure useful uncertainty. The BO method can be greatly improved

in high-dimensional tasks by using trust-region-based search restrictions (Wan et al., 2021; Eriksson et al., 2019; Biswas et al., 2021; Khan et al., 2023) and by combining it with deep generative models (Stanton et al., 2022; Gruver et al., 2024). However, these methods usually suffer from scalability issues due to the complexity of the Gaussian process (GP) kernel (Belanger et al., 2019) or the difficulty of sampling from an intractable posterior (Zhang et al., 2022).

Offline model-based optimization (MBO) (Kumar & Levine, 2020; Trabucco et al., 2021; Yu et al., 2021; Chen et al., 2022; Kim et al., 2023; Chen et al., 2023a; Yun et al., 2024) also addresses the design of biological sequences using offline datasets only, which can be highly efficient because they do not require oracle queries. These approaches have reported meaningful findings, such as the conservative requirements on proxy models since proxy models tend to give high rewards on unseen samples (Trabucco et al., 2021; Yu et al., 2021; Yuan et al., 2023; Chen et al., 2023b). This supports our approach of adaptive conservatism in the search process. Surana et al. (2024) recently noted that offline design and existing benchmarks are insufficient to reflect biological reliability, indicating that settings without additional oracle queries might be too idealistic. For a more comprehensive discussion, see the work by Kim et al. (2025).

Reinforcement learning methods, such as DyNA PPO (Angermueller et al., 2020) and GFlowNets (Bengio et al., 2021; Jain et al., 2022; 2023b; Hernández-García et al., 2024), and sampling with generative models (Brookes & Listgarten, 2018; Brookes et al., 2019; Das et al., 2021; Song & Li, 2023) aim to search the biological sequence space using a sequential decision-making process with a policy, starting from scratch. Similarly, sampling with generative models (Brookes & Listgarten, 2018; Brookes et al., 2019; Song & Li, 2023) searches the sequence space using generative models like VAE (Kingma & Welling, 2014). While these approaches allow for the creation of novel sequences, as sequences are generated from scratch, they are relatively prone to incomplete proxy models, particularly in regions where the proxy is misspecified due to being out-of-distribution.

An alternative line of research is evolutionary search (Arnold, 1998; Bloom & Arnold, 2009; Schreiber et al., 2020; Sinai et al., 2020; Ren et al., 2022; Ghari et al., 2023; Kirjner et al., 2024), a popular method in biological sequence design. These methods iteratively edit sequences and constrain new sequences so as not to deviate too far from the seed sequence; usually the *wild-type*, which occurs in nature. This can be viewed as constrained optimization, where out-of-distribution inputs to the proxy model get avoided, as they can lead to unrealistic and low-score biological sequences. As such, they do not aim to produce highly novel sequences.

## 5.2. GFlowNets

GFlowNets were introduced by Bengio et al. (2021) and unified by Bengio et al. (2023), demonstrating effectiveness across various domains, including language modeling (Hu et al., 2023), diffusion models (Sendera et al., 2024; Venkatraman et al., 2024), and scientific discovery (Jain et al., 2022; 2023a; AI4Science et al., 2023). Several works have aimed to improve their training methods for better credit assignment (Malkin et al., 2022; Madan et al., 2023; Pan et al., 2023; Jang et al., 2024) and extensions to multi-objective settings (Jain et al., 2023b; Chen & Mauch, 2024). Complementary to these, several studies have investigated advanced off-policy exploration strategies (Rector-Brooks et al., 2023; Shen et al., 2023; Kim et al., 2024d;c;a;b; Madan et al., 2025). Our method is particularly related to these exploration methods, yet the major difference is that they are designed under the assumption that the reward model is accurate, which does not hold in active learning and thus requires conservativeness.

**GFlowNets for biological sequence design.** GFlowNets were first introduced for active learning in biological sequence design by Jain et al. (2022), showing that off-policy updates can improve training stability and efficiency by leveraging static annotated datasets. They also incorporated epistemic uncertainty to encourage exploration in a curiosity-driven manner. Separately, Ghari et al. (2023) proposed GFNSeqEditor, which uses GFlowNets as priors for evolutionary editing of biological sequences, rather than for de novo generation. Their flow model is trained offline (i.e., without a proxy model) and used to generate diverse edits for a given input sequence. Our method can be seen as a hybrid of off-policy RL and evolutionary search, capitalizing on both the high novelty offered by GFlowNets and the high rewards with out-of-distribution robustness provided by constrained search, when they are properly balanced by $\delta$. Our experimental results in Section 6.3 demonstrate that $\delta$-CS achieves balanced exploration.

In this paper, we focus on applying $\delta$-CS within GFlowNets. Recent work has established a theoretical equivalence between GFlowNet training objectives, such as detailed balance and trajectory balance, and well-known off-policy maximum entropy RL algorithms, including Soft Q-Learning (Haarnoja et al., 2017) and Path Consistency Learning (Nachum et al., 2017), when appropriate reward corrections are applied (Tiapkin et al., 2024; Deleu et al., 2024). The experimental results are also provided in Appendix B.2.

## 6. Experiments

We first investigate proxy failures and the impact of our conservative search by directly applying $\delta$-CS to GFN-AL (Jain et al., 2022) in Section 6.1. Subsequently, in Sec-

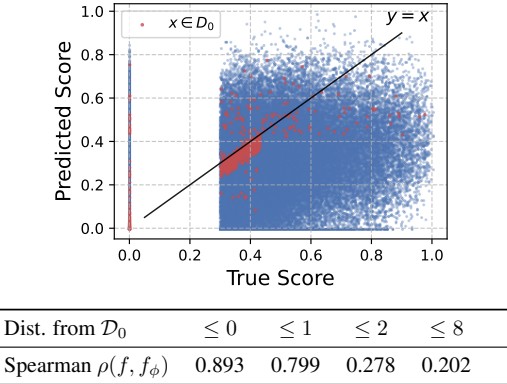

| Dist. from $\mathcal{D}_0$ | $\leq 0$ | $\leq 1$ | $\leq 2$ | $\leq 8$ |
|---|---|---|---|---|
| Spearman $\rho(f, f_\phi)$ | 0.893 | 0.799 | 0.278 | 0.202 |

*Figure 2.* Proxy failure on Hard TF-Bind-8. '$\leq 0$' and '$\leq 8$' denote the initial dataset and the whole sequence space, respectively.

tion 6.2, we evaluate the proposed method on tasks from the FLEXS benchmark (Sinai et al., 2020). Section 6.3 provides a deeper comparison with GFNSeqEditor and GFN-AL, highlighting $\delta$-CS's balanced use of proxy models. Finally, we investigate noise injection and denoising policies by comparing with local search GflowNets (Kim et al., 2024d) in Section 6.4. All results are reported over five independent runs.[2]

### 6.1. Study on proxy failure and the effect of $\delta$-CS

**Task: Hard TF-Bind-8.** We aim to generate DNA sequences (length $L = 8$) that maximize the binding affinity to the target transcription factor. Comprehensive analysis is allowed since the full sequence space is characterized by experiments (Barrera et al., 2016). However, the original task, especially due to the large initial dataset, is easy to optimize (Sinai et al., 2020), making performance differences among methods less clear. To clearly verify the effect of $\delta$-CS, we introduce a hard variant by modifying both the initial dataset distribution and the scoring landscape. Specifically, we collect the initial dataset near a certain sequence while ensuring that the initial sequences have lower scores than the given sequence to form a dataset $\mathcal{D}_0$ of size 1,024. We further modify the landscape to assign zero to any sequence scoring below 0.3. These modifications better reflect realworld design scenarios where the search space is vast, data is limited, and many sequences can have near-zero scores.

**Proxy failure.** Figure 2 illustrates the proxy values and true scores for all $x \in \mathcal{X}$ in the first round. While the proxy provides accurate predictions for the initial data points $x \in \mathcal{D}_0$ (represented by the red dots), it produces unreliable predictions for points outside $\mathcal{D}_0$. The correlation between $f$ and $f_\phi$ significantly decreases as data points move farther from the observed initial dataset (denoted as '$\leq 0$' in the

---

[2]Available at https://github.com/hyeonahkimm/delta_cs.

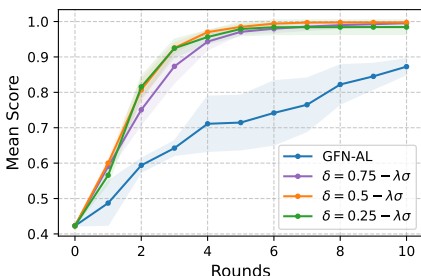

*Figure 3.* Mean score over rounds on Hard TF-Bind-8.

table). This supports our hypothesis that the proxy model performs poorly on out-of-distribution data.

**Effect of $\delta$ conservativeness.** As shown in Figure 3, $\delta$-CS with $\delta < 1$ significantly improves GFN-AL by focusing on sequences that maintain higher correlation with the ground-truth scores. By restricting the search space to a smaller edit distance, $\delta$-CS better aligns with ground-truth scores. By limiting the search within these constrained edit distances, $\delta$-CS enhances the correlation with the oracle. Even with a fixed $\delta$, Figure 9 in Appendix B.6 shows consistent gains. We also confirm similar improvements on the anti-microbial peptide design (AMP) task from Jain et al. (2022); detailed results in Appendix B.1 show that $\delta$-CS consistently outperforms the original GFN-AL.

### 6.2. Experiments on FLEXS Benchmark

**Implementation details.** For proxy models, we employ a convolutional neural network with one-dimensional convolutions according to (Sinai et al., 2020). We use a UCB acquisition function and measure the uncertainty with an ensemble of three network instances. We use $\delta_{\text{const}} = 0.5$ for DNA ($L = 8$) and RNA ($L = 14$) sequence design and $\delta = 0.05$ for protein design ($L = 238, 90$). Lastly, we set $\lambda$ to satisfy $\lambda \mathbb{E}_{\mathcal{D}_0(x)} \sigma(x) \approx \frac{1}{L}$ based on the observations from the initial round. More details are provided in Appendix A.1 and Appendix A.2.

**Baselines.** As our baseline methods, we employ representative exploration algorithms. We use the same architecture to implement proxy models for all baselines except for GFN-AL, where we adopt the original implementation. Further details are provided in Appendix A.3.

- **AdaLead** (Sinai et al., 2020) is a model-guided evaluation method with a hill-climbing algorithm.
- **Bayesian optimization** (BO; Snoek et al., 2012) is a classical algorithm for black-box optimization. We employ the BO algorithm with a sparse sampling of the mutation space implemented by Sinai et al. (2020).
- **TuRBO** (Eriksson et al., 2019) is a Bayesian optimization algorithm that partitions search space into promising local regions and performs the optimization process within the region.

- **CMA-ES** (Hansen, 2006) is a well-known evolutionary algorithm that optimizes a continuous relaxation of one-hot vectors encoding sequence with the covariance matrix.
- **CbAS** (Brookes et al., 2019) and **DbAS** (Brookes & List-garten, 2018) are probabilistic frameworks that use model-based adaptive sampling with a variational autoencoder (VAE; Kingma & Welling, 2014). Notably, CbAS restricts the search space with a trust-region search similar to the proposed method.
- **DyNA PPO** (Angermueller et al., 2020) uses proximal policy optimization (PPO; Schulman et al., 2017), an on-policy training method.
- **GFN-AL** (Jain et al., 2022) is our main baseline that uses GFN with Bayesian active learning.

**Experiment setup.** For each task, we conduct 10 active learning rounds starting from the initial dataset $\mathcal{D}_0$. The query batch size is all set as 128. Further details of each task are provided in the following subsections. To evaluate the performance, we measure the maximum, median, and mean scores of Top-$K$ sequences.

#### 6.2.1. RNA SEQUENCE DESIGN

**Task.** The goal is to design an RNA sequence that binds to the target with the lowest binding energy, which is measured by ViennaRNA (Lorenz et al., 2011). The length ($L$) of RNA is set to 14, with 4 tokens. In this paper, we have three RNA binding tasks, RNA-A, RNA-B, and RNA-C, whose initial datasets consist of 5,000 randomly generated sequences with certain thresholds; we adopt the offline dataset provided in Kim et al. (2023). We use $\delta = 0.5$ and $\lambda = 5$.

**Results.** As shown in Figure 4, our method outperforms all baseline approaches. The curve in Figure 4 increases significantly faster than the other methods, indicating that $\delta$-CS effectively trains the policy and generates appropriate queries in each active round. See Appendix C.1 in details.

#### 6.2.2. DNA SEQUENCE DESIGN

**Task.** In this task, we follow the setup from the previous studies (Sinai et al., 2020; Jain et al., 2022; Trabucco et al., 2022; Kim et al., 2023). The initial dataset $\mathcal{D}_0$ is the bottom 50% in terms of the score, which results in $32,898$ samples, with a maximum score of 0.439. Similar to RNA, we use $\delta = 0.5$ and $\lambda = 5$.

**Results.** In TF-Bind-8, every method successively discovers DNA sequences with high binding affinity scores in the early rounds. Nevertheless, the results show that $\delta$-CS further improves GFN-AL by restricting search space; see the max, median, diversity, and novelty in Appendix C.2. In addition, $\delta$-CS demonstrates the best median scores, as well.

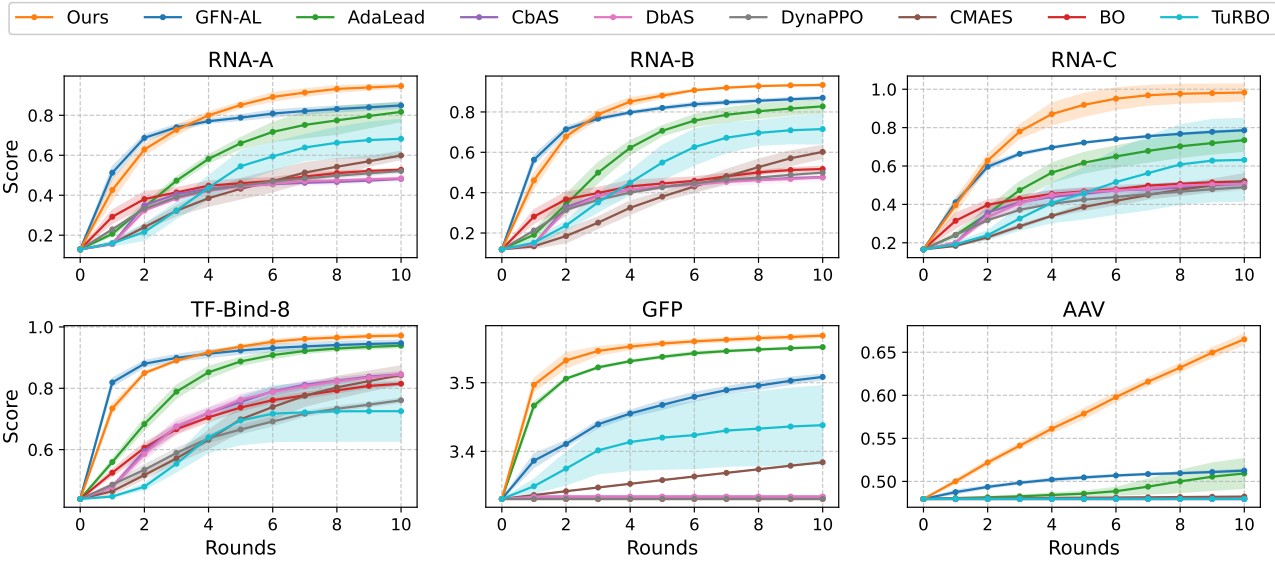

*Figure 4.* Mean scores of Top-128 over active rounds. Ours (GFN-AL + $\delta$-CS) consistently outperforms baseline in RNA, DNA (TF-Bind-8), and protein (GFP and AAV) design tasks.

### 6.2.3. PROTEIN SEQUENCE DESIGN

We consider two protein sequence design tasks: the green fluorescent protein (GFP; Sarkisyan et al., 2016) and additive adeno-associated virus (AAV; Ogden et al., 2019).

**GFP.** The objective is to identify protein sequences with high log-fluorescence intensity values.[3] The vocabulary is defined as 20 standard amino acids, i.e., $|\mathcal{V}| = 20$, and the sequence length $L$ is 238; thus, we set $\delta$ as 0.05 and $\lambda$ as 0.1, according to our guideline. The initial datasets are generated by randomly mutating the provided wild-type sequence for each task while filtering out sequences that have higher scores than the wild-type; we obtain the initial dataset with $|\mathcal{D}_0| = 10\,200$ with a maximum score of 3.572.

**AAV.** The aim is to discover sequences that lead to higher gene therapeutic efficiency. The sequences are composed of the 20 standard amino acids with a length of 90, resulting in the search space of $20^{90}$. In the same way as in GFP, we collect an initial dataset of 15,307 sequences with a maximum score of 0.500. We use $\delta = 0.05$ and $\lambda = 1$.

**Results.** Figure 4 shows the results of all methods in protein sequence design tasks. Given the combinatorially vast design space with sequence lengths $L = 238$ and 90, most baselines fail to discover new sequences whose score is higher than the maximum of the dataset. In contrast, our method finds high-score sequences beyond the dataset, even with a single active round. This underscores the superior-

ity of our search strategy in practical biological sequence design tasks. Full results are provided in Appendix C.3.

### 6.3. Achieving Pareto Frontier with Balanced Conservativeness

In this analysis, we demonstrate that $\delta$-CS achieves a balanced search using $\delta$, producing Pareto frontiers or comparable results to the baseline methods: GFN-AL (Jain et al., 2022) and GFNSeqEditor (Ghari et al., 2023). Notably, GFN-AL can be seen as a variant of our method with $\delta = 1$, which fully utilizes the entire trajectory search. This approach is expected to yield high novelty and diversity, but it is also prone to generating low rewards due to the increased risk of out-of-distribution samples affecting the proxy model. GFNSeqEditor, on the other hand, leverages GFlowNets as a prior, editing from a wild-type sequence. It is designed to deliver reliable performance and be more robust to out-of-distribution issues by constraining the search to sequences similar to the wild type. However, unlike $\delta$-CS, GFNSeqEditor does not utilize such obtained samples for training GFlowNets in full trajectory level; GFNSeqEditor is expected to have lower diversity and novelty compared to GFN-AL and $\delta$-CS.

As shown in Figure 5, GFN-AL generally produces higher diversity and novelty in the RNA and GFP tasks compared to GFNSeqEditor. However, GFNSeqEditor performs better in terms of reward on the large-scale GFP task, whereas GFN-AL struggles due to the lack of a constrained search procedure in such a large combinatorial space. In contrast, $\delta$-CS achieves Pareto-optimal performance compared to both

---

[3]FLEXS uses TAPE (Rao et al., 2019), whereas Jain et al. (2022) employs the Design-Bench Transformer (Trabucco et al., 2022). Additional results are provided in Appendix B.5.

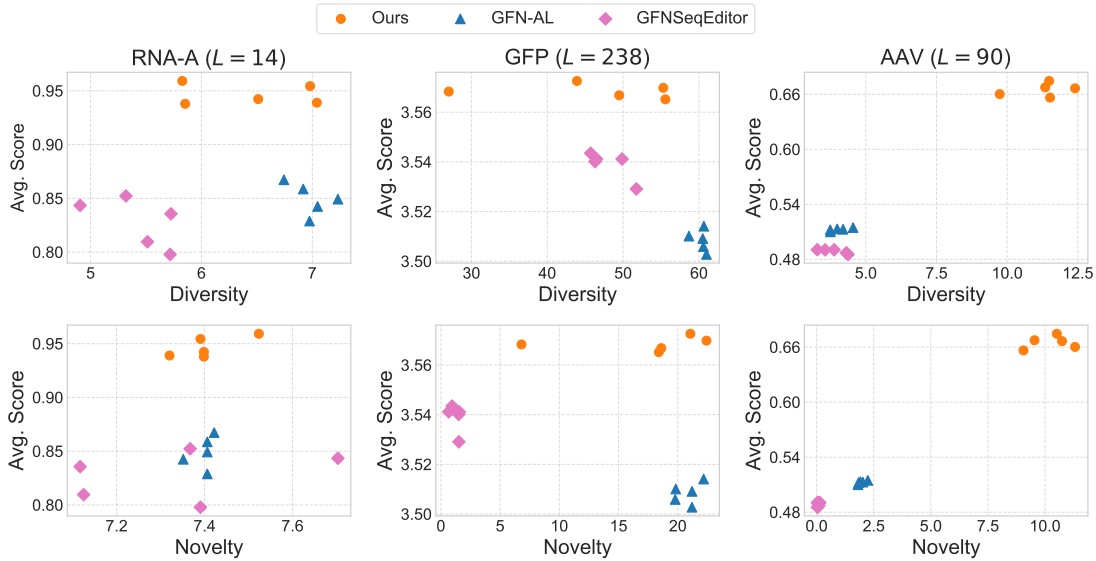

*Figure 5.* Average score and diversity/novelty with five independent runs. Our method (GFN-AL + $\delta$-CS) consistently approaches Pareto frontier performance. We set $\delta = 0.5$ for short sequences ($L \leq 50$) and set $\delta = 0.05$ for long length sequences ($L > 50$).

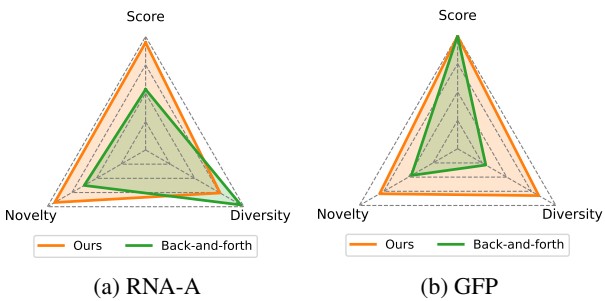

(a) RNA-A       (b) GFP

*Figure 6.* Comparison $\delta$-CS with back-and-forth search in LS-GFN. Further results are provided in Appendix B.3

methods, clearly outperforming GFNSeqEditor across six tasks, with higher rewards, diversity, and novelty. For the RNA and GFP tasks, we achieve higher scores than GFN-AL while maintaining similar novelty but slightly lower diversity. In the AAV task, $\delta$-CS shows a distinct Pareto improvement. These results demonstrate that $\delta$-CS provides a beneficial balance by combining conservative search with amortized inference on full trajectories using off-policy GFlowNets training, effectively capturing the strengths of both GFN-AL and GFNSeqEditor. The results with various $\delta$ are provided in Appendix B.8.

### 6.4. Comparison with Back-and-Forth Search

The noise injection and denoising policy shares similarities with the back-and-forth strategy suggested in local search GFlowNets (LS-GFN; Kim et al., 2024d). LS-GFN uses a backward policy to partially destroy a solution and a forward policy to reconstruct a new solution. While LS-GFN

does not inherently use proxy models, we adapt it for a conservative search comparison by setting its backtracking step $K$ to $\delta L$, aligning its level of conservativeness with ours.

As illustrated in Figure 6, our method achieves more balanced trade-offs among high scores, diversity, and novelty than LS-GFN, likely due to its flexibility in selecting which tokens to mask. In an auto-regressive sequence generation setting, the backtracking in LS-GFN destroys only the last tokens, limiting the local search region, whereas our approach can randomly mask any token in a sequence. Moreover, as shown in Table 3 and Figure 8, our approach demonstrates it remains more robust when the initial dataset is restricted to lower-quality sequences; see Appendix B.4.

### 6.5. Further Studies

**Studies on the effect of $\delta$-CS with various off-policy RL.**

**Studies on $\delta$.** We study the choice of $\delta_{\mathrm{const}}$ and the effectiveness of the adaptive $\delta(x, \sigma)$. In Appendix B.7 and Appendix B.8, we investigate how varying $\delta_{\mathrm{const}}$ affects performance. While $\delta$-CS generally outperforms GFN-AL, its gains diminish as $\delta_{\mathrm{const}}$ grows, because $\delta = 1$ reverts to on-policy exploration without any conservative constraint. Furthermore, the results in Appendix B.6 verify the benefit of adaptive $\delta(x, \sigma)$. While even a fixed $\delta$ improves GFN-AL on Hard TF-Bind-8, adaptive $\delta$ yields additional performance gains in overall.

**Studies on the varying proxy qualities.** We further validate $\delta$-CS under degraded proxy models by truncating the initial dataset to its 50%, 25%, and 10% score percentiles.

As shown in Figure 8, across all truncation levels, $\delta$-CS consistently outperforms GFN-AL and LS-GFN, confirming that incorporating conservative search is crucial for robustness against proxy misspecification; the details are provided in Appendix B.4.

## 7. Conclusions and discussion

In this paper, we introduced a novel off-policy sampling method for GFlowNets, called $\delta$-CS, which provides controllable conservativeness through the use of a $\delta$ parameter. Additionally, we proposed an adaptive conservativeness approach by adjusting $\delta$ for each data point based on prediction uncertainty. We demonstrated the effectiveness of $\delta$-CS in active learning GFlowNets, achieving strong performance across various biological sequence design tasks, including DNA, RNA, protein, and peptide design, consistently outperforming existing baselines.

**Limitations and future works.** The main limitation of our method is that it does not fundamentally resolve the drawbacks of active learning; it serves as a useful tool within the existing framework. Investigating robust proxy training and uncertainty measurement techniques remains necessary. These improvements are orthogonal to our approach and can enhance $\delta$-CS when integrated. Future work includes combining $\delta$-CS with existing GFlowNet methods. For example, applying improved credit assignment for larger-scale tasks (Jang et al., 2024) and extending to multi-objective settings (Jain et al., 2023b; Chen & Mauch, 2024) could significantly boost its applicability and effectiveness.

## Impact Statement

This work tackles a common challenge in scientific discovery: generating candidates with limited data and unreliable proxy models. In biological sequence design, we introduce a conservative search strategy that adapts exploration range based on proxy uncertainty to reduce reward hacking. While developed for this domain, the principle of aligning exploration with model confidence may generalize to other domains where generative models and active learning pipelines face similar constraints.

## Acknowledgements

Minsu Kim is supported by KAIST Jang Yeong Sil Fellowship and the Canadian AI Safety Institute Research Program at CIFAR through a Catalyst award. This work was supported by the National Research Foundation of Korea (NRF) grant funded by the Korean government (MSIT) (RS-2025-00563763).

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

# A. Implementation detail

---
**Algorithm 1** Active Learning GFlowNets with $\delta$-CS
---

1: **Input:** oracle $f$, initial dataset $\mathcal{D}_0$, active rounds $T$, query size $B$, training batch size $2 \times M$.
2: **for** $t = 1$ **to** $T$ **do**
3:    ▷   **STEP A:** PROXY TRAINING
4:    **for** $k = 1$ **to** `numProxyTrain` **do**
5:      train proxy $f_\phi(x)$ with current round dataset $\mathcal{D}_{t-1}$: $\mathcal{L}(\phi) = \mathbb{E}_{x \sim P_{\mathcal{D}_{t-1}}(x)}\left[ (f(x) - f_\phi(x))^2 \right]$.
6:    **end for**
7:    ▷   **STEP B:** POLICY TRAINING
8:    **for** $k = 1$ **to** `numPolicyTrain` **do**
9:      obtain off-policy trajectories $\hat{\tau}_1, \ldots, \hat{\tau}_M$ from $\hat{x}_1, \ldots, \hat{x}_M$ given by $\delta$-CS $(\mathcal{D}_{t-1}, M, \delta)$.
10:     obtain offline trajectory $\tau_1, \ldots, \tau_M$ from $x_1, \ldots, x_M \sim P_{\mathcal{D}_{t-1}}(\tau)$.
11:     train $\theta$ with TB loss over $\hat{\tau}_1, \ldots, \hat{\tau}_M$ and $\tau_1, \ldots, \tau_M$

$$\mathcal{L}(\theta) = \frac{1}{2M} \sum_{i=1}^{M} \left( \log \frac{Z_\theta P_F(\tau_i; \theta)}{R(x_i; \phi)} \right)^2 + \frac{1}{2M} \sum_{i=1}^{M} \left( \log \frac{Z_\theta P_F(\hat{\tau}_i; \theta)}{R(\hat{x}_i; \phi)} \right)^2.$$

12:    **end for**
13:    ▷   **STEP C:** DATASET AUGMENTATION WITH ORACLE $f$ QUERY
14:    obtain query samples $\hat{x}_1, \ldots, \hat{x}_B$ from $\delta$-CS $(\mathcal{D}_{t-1}, B, \delta)$.
15:    $\mathcal{D}_t \leftarrow \mathcal{D}_{t-1} \cup \{(\hat{x}_i, f(\hat{x}_i))\}_{i=1}^{B}$.
16: **end for**

---
**Algorithm 2** Sampling with $\delta$-CS
---

1: **Input:** Dataset $\mathcal{D}_{t-1}$, batch size $M$, conservative parameter $\delta$
2: sample high reward data $x_1, \ldots, x_M$ with **rank-based reweighed prior** $P_{\mathcal{D}_{t-1}}(\cdot)$.
3: obtain masked data $\tilde{x}_1, \ldots, \tilde{x}_M$ with **noise injection policy** $P_{\text{noise}}(\tilde{\cdot} | \cdot, \delta)$ from $x_1, \ldots, x_M$.
4: obtain denoised $\hat{x}_1, \ldots, \hat{x}_M$ with **denoising policy** $P_{\text{denoise}}(\hat{\cdot} | \tilde{\cdot}; \theta)$ from $\tilde{x}_1, \ldots, \tilde{x}_M$.
5: **return** $\hat{x}_1, \ldots, \hat{x}_M$.

---

## A.1. Proxy training

For training proxy models, we follow the procedure of (Jain et al., 2022). We use Adam (Kingma, 2015) optimizer with learning rate $1 \times 10^{-5}$ and batch size of 256. The maximum proxy update is set as 3000. To prevent over-fitting, we use early stopping using the $10\%$ of the dataset as a validation set and terminate the training procedure if validation loss does not improve for five consecutive iterations.

## A.2. Policy training

As described in Section 6, we employ a two-layer long short-term memory (LSTM; Hochreiter & Schmidhuber, 1997) with 512 hidden dimensions. The policy is trained with a learning rate of $5 \times 10^{-4}$ with a batch size of 256. The learning rate of $Z$ is set as $10^{-3}$. The coefficient $\kappa$ in Section 3.2 is set as 0.1 for TF-Bind-8 and AMP with MC dropout, according to Jain et al. (2022), and 1.0 for RNA and protein design with Ensemble following Ren et al. (2022).

## A.3. Implementation details of baselines

We adopt open-source code from FLEXS benchmark (Sinai et al., 2020).

- **AdaLead** (Sinai et al., 2020): We use a default settings of hyperparmeters for AdaLead. Specifically, we use a recombination rate of 0.2, mutation rate of $1/L$, where $L$ is sequence length, and threshold $\tau = 0.05$.

- **DbAS** (Brookes & Listgarten, 2018): We implement DbAS with variational autoencoder (VAE; Kingma & Welling, 2014) as the generator. The input is a one-hot encoding vector, and the output latent dimension is 2. In each cycle, DbAS starts by training the VAE with the top 20% sequences in terms of the score.

- **CbAS** (Brookes et al., 2019): Similar to DbAS, we implement CbAS with VAE. The main difference from DbAS is that we select top 20% sequences with the weights $p(\boldsymbol{x}|\boldsymbol{z}, \theta^{(0)})/q(\boldsymbol{x}|\boldsymbol{z}, \phi^{(t)})$, where $p(\cdot; \theta^{(0)})$ is trained with the ground-truth samples and $q(\cdot; \phi^{(t)})$ is trained on the generated sequences over $t$ training rounds.

- **DyNA PPO** (Angermueller et al., 2020): We closely follow the algorithm presented in (Angermueller et al., 2020). For a fair comparison, we use CNN ensembles to parameterize the proxy model instead of suggested architectures.

- **CMA-ES** (Hansen, 2006): We implement a covariance matrix adaptation evolution strategy (CMA-ES) for sequence generation. As the generated samples from CMA-ES are continuous, we convert the continuous vectors into one-hot representation by computing the argmax at each sequence position.

- **BO** (Snoek et al., 2012): We use classical GP-BO algorithm for all tasks. For Gaussian Process Regressor (GPR), we use a default setting from the `sklearn` library. For the acquisition function, they use Thompson sampling (Russo et al., 2018).

- **TuRBO** (Eriksson et al., 2019): We closely follow the algorithm presented in (Eriksson et al., 2019). As it supports continuous search space, convert the continuous vectors into one-hot representation by computing the argmax at each sequence position.

Furthermore, we employ GFN-AL and GFNSeqEditor. We adopt the original implementation and setup for TF-Bind-8 and AMP. For newly added tasks, we report better results among the original MLP policy and the LSTM policy. Note that GFP in FLEXS is different from the one employed in GFA-AL; we treat this as a new task based on the observation in the work from Surana et al. (2024).

- **GFN-AL** (Jain et al., 2022): We strictly follow hyperparameters of the original code in they conduct experiments on TF-Bind-8 and AMP. The proxy is parameterized using an MLP with two layers of 2,048 hidden. For the policy, a 2-layer MLP with 2,048 hidden dimensions is used, but we also test it with a 2-layer LSTM policy.

- **GFNSeqEditor** (Ghari et al., 2023): We implemented the editing procedure on top of the GFN-AL. Note that GFNSeqEditor does not utilize the proxy model, so the GFlowNets policy is trained using offline data only with the same policy training procedure of GFN-AL. GFNSeqEditor can also implicitly control the edit percentage with its hyperparameters, which are set $\delta = 0.01, \sigma = 0.0001, \lambda = 0.1$ in this study. Note that $\delta$ is not the conservativeness parameter.

# B. Further studies

## B.1. Anti-microbial peptide design

**Task setup.** The goal is to generate protein sequences with anti-microbial properties (AMP). The vocabulary size $|\mathcal{V}| = 20$, and the sequence length ($L$) varies across sequences, and we consider sequences of length 50 or lower. For the AMP task, we consider a much larger query batch size for each active round because they can be efficiently synthesized and evaluated (Jain et al., 2022). We set $\delta$ as 0.5 with $\lambda = 1$.

*Table 1.* Results on AMP with different acquisition functions (UCB, EI). The mean and standard deviation from five runs are reported. Improved results with $\delta$-CS over GFN-AL are marked in **bold**.

| | Max | Mean | Diversity | Novelty |
|---|---|---|---|---|
| COMs | $0.930 \pm 0.001$ | $0.920 \pm 0.000$ | $0.000 \pm 0.000$ | $11.869 \pm 0.298$ |
| DyNA PPO | $0.953 \pm 0.005$ | $0.941 \pm 0.012$ | $15.186 \pm 5.109$ | $16.556 \pm 3.653$ |
| GFN-AL (UCB) | $0.936 \pm 0.004$ | $0.919 \pm 0.005$ | $\mathbf{28.504 \pm 2.691}$ | $19.220 \pm 1.369$ |
| GFN-AL + $\delta$-CS (UCB) | $\mathbf{0.948 \pm 0.015}$ | $\mathbf{0.938 \pm 0.016}$ | $25.379 \pm 3.735$ | $\mathbf{23.551 \pm 1.290}$ |
| GFN-AL (EI) | $0.950 \pm 0.002$ | $0.940 \pm 0.003$ | $15.576 \pm 7.896$ | $21.810 \pm 4.165$ |
| GFN-AL + $\delta$-CS (EI) | $\mathbf{0.962 \pm 0.003}$ | $\mathbf{0.958 \pm 0.004}$ | $\mathbf{16.631 \pm 2.135}$ | $\mathbf{24.946 \pm 4.246}$ |

**Results.** The results in Table 1 illustrate that ours consistently gives improved performance over GFN-AL regardless of acquisition function. According to the work from Jain et al. (2022), we also adopt conservative model-based optimization method, (COMs; Trabucco et al., 2021) and on-policy reinforcement learning, DyNA PPO (Angermueller et al., 2020) as baselines. Our method demonstrated significantly higher performance in terms of mean, diversity, and novelty compared to the baselines.

## B.2. $\delta$-CS with various off-policy RL

We extend our experiments on protein design tasks to include three soft off-policy RL algorithms: Soft Q-Learning (SQL) (Haarnoja et al., 2017), Log-Partition Variance Gradient (VarGrad; Richter et al., 2020), and Soft Path Consistency Learning (Soft PCL) (Nachum et al., 2017; Chow et al., 2018; Madan et al., 2023), yielding four baselines in total (TB, SQL, VarGrad, and Soft PCL). Trajectory Balance (TB) and VarGrad apply constraints over full trajectories, resulting in higher variance but lower bias. TB explicitly estimates state flows or terminal partition functions, whereas VarGrad does so implicitly via batch averaging. Soft PCL operates over sub-trajectories (akin to TD-$\lambda$), balancing bias and variance through trajectory length. SQL relies on one-step transitions, reducing variance but increasing bias. For a broader discussion on connections between off-policy RL in discrete sampling, see the works by Tiapkin et al. (2024) and Deleu et al. (2024).

*Table 2.* Mean score of Top-128 after ten active rounds

|  | GFP | AAV |
|---|---|---|
| SQL | $3.331 \pm 0.000$ | $0.480 \pm 0.000$ |
| SQL + $\delta$-CS | $3.573 \pm 0.003$ | $0.495 \pm 0.001$ |
| VarGrad | $3.331 \pm 0.000$ | $0.480 \pm 0.000$ |
| VarGrad + $\delta$-CS | $3.567 \pm 0.003$ | $0.668 \pm 0.011$ |
| Soft PCL ($\lambda$=0.9) | $3.331 \pm 0.000$ | $0.480 \pm 0.000$ |
| Soft PCL ($\lambda$=0.9) + $\delta$-CS | $3.578 \pm 0.004$ | $0.622 \pm 0.011$ |

As shown in Table 2, $\delta$-CS consistently improves performance across all methods, demonstrating its plug-and-play versatility. Since the global sequence structure is crucial in protein design, full-trajectory methods (TB and VarGrad with $\delta$-CS) tend to outperform shorter-horizon approaches.

### B.3. Comparison with back-and-forth search in LS-GFN

Our search with noise injection and denoising policies shares similar insight with a back-and-forth search in local search GFlowNets (LS-GFN; Kim et al., 2024d). They both induce new sequences by partially revising given sequences. In particular, LS-GFN incorporates a back-and-forth search strategy that partially backtracks trajectories using a backward policy and reconstructs them using a forward policy of the GFN. This section compares the search strategies under the conservative search; we control the conservativeness of LS-GFN by setting the backtracking step $K$ as $\delta L$.

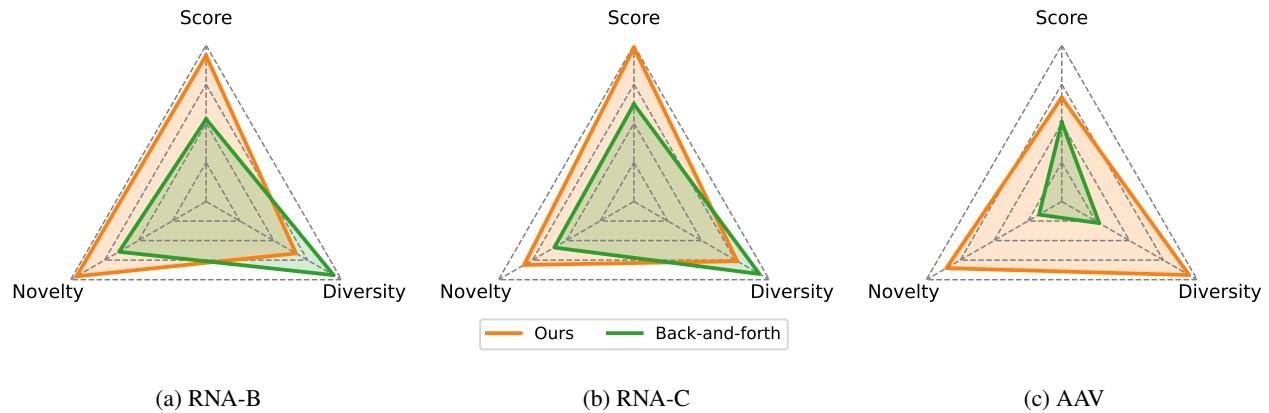

(a) RNA-B            (b) RNA-C            (c) AAV

*Figure 7.* Comparison $\delta$-CS with back-and-forth search in LS-GFN on RNA-B,C and AAV

## B.4. Studies with smaller initial dataset on protein designs

In this section, we extend our ablation study by comparing our method, $\delta$-CS, against two baselines: GFN-AL and an additional off-policy search method, LS-GFN (Kim et al., 2024d). This study evaluates the performance of $\delta$-CS under varying initial dataset sizes, $|\mathcal{D}_0| = 1,000$, and compares the results to the original ablation study setup.

As shown in Table 3, our $\delta$-CS demonstrates a substantial advantage over both GFN-AL and its improved variant, LS-GFN, which leverages back-and-forth search. The results highlight the effectiveness of $\delta$-CS in enhancing GFN training by enabling a more robust and conservative off-policy search, which is critical for improving proxy-based active learning.

*Table 3.* Ablation study results with 1,000 initial datapoints for GFP and AAV tasks, showing maximum values achieved after active learning.

| Method | GFP | AAV |
|---|---|---|
| Adalead | $3.568 \pm 0.005$ | $0.557 \pm 0.023$ |
| GFN-AL | $3.586 \pm 0.006$ | $0.560 \pm 0.008$ |
| GFN-AL + LS (Kim et al., 2024d) | $3.580 \pm 0.003$ | $0.493 \pm 0.006$ |
| **GFN-AL + $\delta$-CS** | $\mathbf{3.591 \pm 0.007}$ | $\mathbf{0.704 \pm 0.024}$ |

To further verify that $\delta$-CS is robust to proxy misspecification compared to other GFN methods, we conducted additional experiments to test whether this hypothesis holds at different levels of proxy model quality.

To degrade the proxy model quality, we truncated the initial dataset at different levels—$50\%$, $25\%$, and $10\%$ percentiles based on reward values. Proxy models trained on datasets with lower percentile cutoffs are more misspecified for higher-reward data points, making GFN training and search more challenging. Under these circumstances, we compared our method with GFN-AL and LS-GFN (Kim et al., 2024d) as GFN baselines.

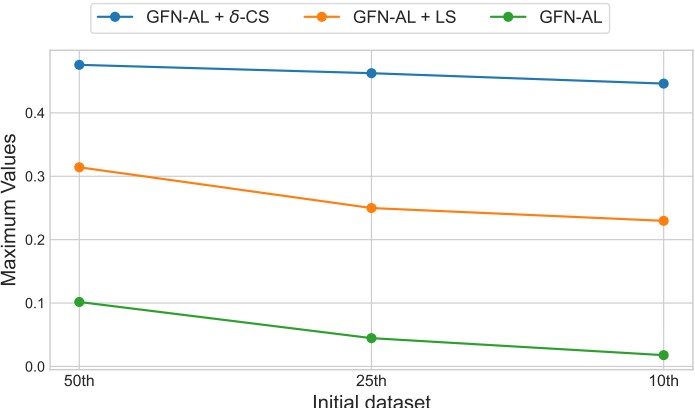

*Figure 8.* Maximum values achieved after active learning with varying initial dataset quality (AAV task).

As shown in Figure 8, the performance decreases as the percentile decreases, which is expected because the proxy quality deteriorates significantly. Among the baselines, our method consistently provides substantially better performance than the others. This demonstrates that our hypothesis—that a conservative search with $\delta$-CS is necessary—holds across different levels of proxy model quality.

### B.5. Experiments on different GFP benchmark

The GFP task has multiple benchmark variants, primarily due to differences in oracle models (e.g., TAPE (Rao et al., 2019), Design-Bench Transformer (Trabucco et al., 2022)) and scoring normalization strategies. Some studies use normalized oracle scores (Kirjner et al., 2024; Lee et al., 2024), while others adopt raw scores (Sinai et al., 2020; Jain et al., 2022; Ren et al., 2022); see the work by Surana et al. (2024) for a comprehensive comparison.

To enable consistent comparison across benchmarks, we follow the experimental setup of LatProtRL (Lee et al., 2024). Specifically, we run 15 active learning rounds, evaluating 256 sequences per round, using the oracle model and initial dataset split as in the work by Kirjner et al. (2024). Following LatProtRL, we report the median top-128 score across 5 independent runs. Results for AdaLead and LatProtRL are directly obtained from the work by Lee et al. (2024).

*Table 4.* GFP optimization results on the benchmark from Kirjner et al. (2024) and Lee et al. (2024)

|  | GFP Medium | GFP Hard |
| --- | --- | --- |
| AdaLead | $0.93 \pm 0.0$ | $0.75 \pm 0.1$ |
| LatProtRL | $0.93 \pm 0.0$ | $0.85 \pm 0.0$ |
| GFN-AL ($\delta = 1$) | $0.21 \pm 0.0$ | $0.09 \pm 0.0$ |
| GFN-AL + $\delta$-CS ($\delta = 0.05$) | $0.57 \pm 0.0$ | $0.60 \pm 0.1$ |
| GFN-AL + $\delta$-CS ($\delta = 0.01$) | $1.06 \pm 0.1$ | $0.86 \pm 0.0$ |

As shown in Table 9, $\delta$-CS consistently outperforms baseline methods across all benchmark variants. In particular, it significantly improves upon GFN-AL (equivalent to $\delta = 1$, i.e., without conservative search), achieving the best results with $\delta = 0.01$. This benchmark is intentionally designed to be more challenging compared to Design-Bench, as the initial datasets in GFP-medium and GFP-hard consist of low-scoring and distant sequences to the 99th percentile sequences; see Appendix A in the work by Kirjner et al. (2024). This results in greater proxy uncertainty, which in turn necessitates a lower $\delta$ to achieve effective exploration.

## B.6. Studies on effect of adaptive $\delta$

This section provides the experimental results to verify the effectiveness of considering uncertainty on each data point for adjusting the conservativeness.

### B.6.1. HARD TF-BIND-8

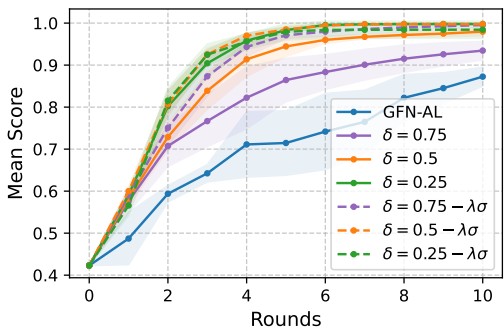

*Figure 9.* Median score over rounds on Hard TF-Bind-8.

To verify its effectiveness and give intuition about how to set $\delta$, we conduct experiments with various $\delta$ in Hard TF-Bind-8. The results show that $\delta$-CS with $\delta < 1$ can significantly outperform GFN-AL by searching for data points that correlate better with the oracle. This holds even when we use constant values for $\delta$. However, as depicted in Figure 9, using adaptive $\delta(x, \sigma)$ mostly gives the improved scores.

### B.6.2. RNA SEQUENCE DESIGN

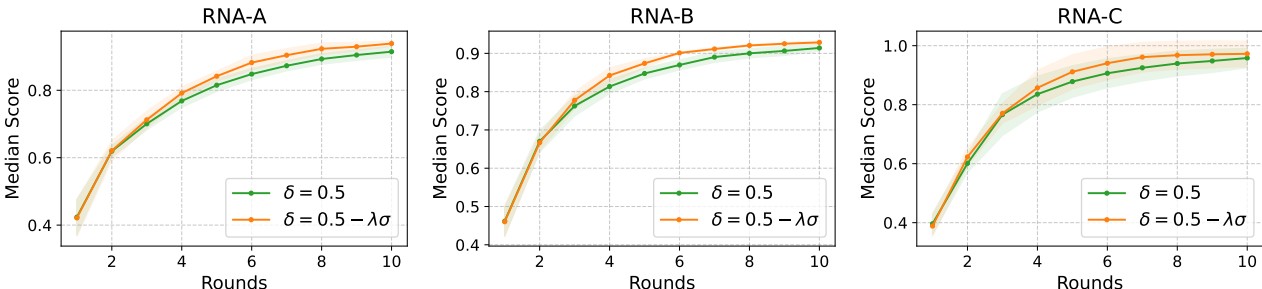

*Figure 10.* Effect of adaptive control on RNA

We examine the effects of proxy uncertainty-based $\delta$. In RNA, the average proxy standard deviation at the initial round is observed as 0.005 to 0.012. Therefore, we set $\lambda = 5$ to roughly make $\lambda\bar{\sigma} \approx 1/L$, where $L = 14$. As illustrated in Figure 10, $\delta(x; \sigma)$ consistently gives the higher score. However, the constant $\delta = 5$ still outperforms all baselines, exhibiting the robustness of $\delta$-CS.

### B.7. Sensitivity analysis over various $\delta_{\text{const}}$.

This section provides the experimental results with varying $\delta_{\text{const}}$ on RNA and protein designs. Note that we use $\delta_{\text{const}} = 0.5$ in the main experiments, meaning that approximately 7 tokens are masked in RNA designs, while we use $\delta_{\text{const}} = 0.05$ for protein designs–approximately masking 4-12 tokens for GFP ($L = 238$) and AAV ($L - 90$).

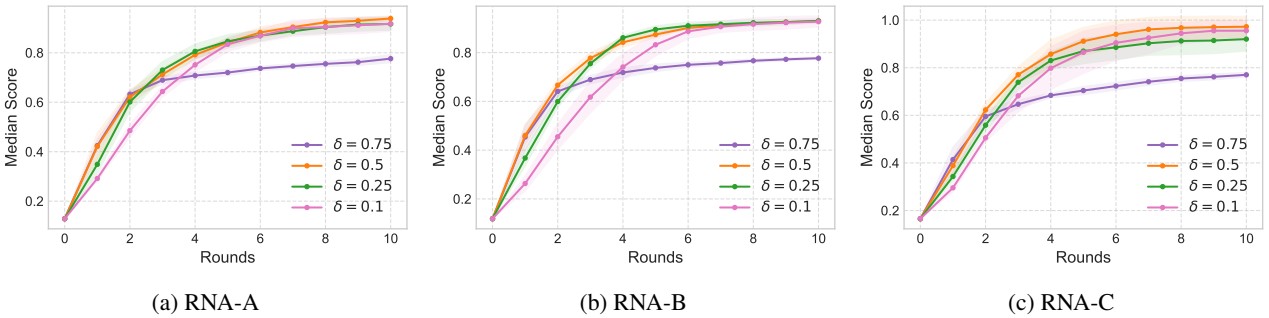

*Figure 11.* Adaptive delta with various $\delta_{\text{const}}$ on RNA

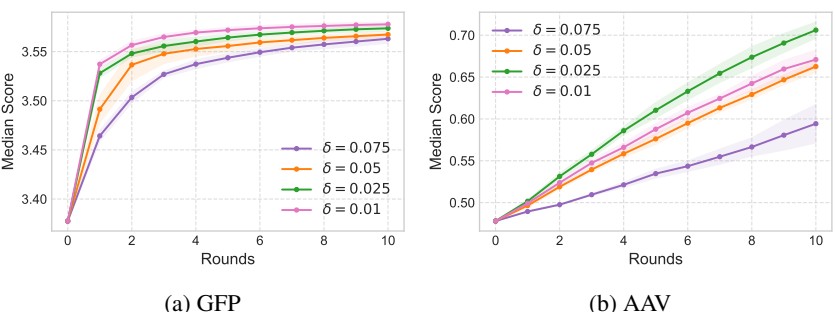

*Figure 12.* Adaptive delta with various $\delta_{\text{const}}$ on protein designs

The results in Figure 11 and Figure 12 show that GFN-AL with $\delta$-CS consistently gives improved results compared to GFN-AL. Even in AAV, $\delta_{\text{const}} = 0.075$ outperforms other baseline, including GFN-AL whose score is 0.509. Moreover, we obtain even better results by setting $\delta_{\text{const}} = 0.01$ in GFP and $\delta_{\text{const}} = 0.025$ in AAV, indicating that domain-specific tuning of $\delta_{\text{const}}$ can yield further gains.

## B.8. Balancing capability with various $\delta$

Similar to Section 6.3, we evaluate $\delta$-CS on RNA-B and RNA-C by varying $\delta_{\text{const}}$ from 0.1 to 0.5. As illustrated in Figure 13, $\delta$-CS consistently reaches the Pareto frontier. In the protein designs, we vary $\delta_{\text{const}}$ between 0.01 and 0.05. Figure 14 demonstrates that GFN-AL with $\delta$-CS forms the Pareto curve over both GFN-AL and GFNSeqEditor, further confirming the balancing capability of $\delta$-CS.

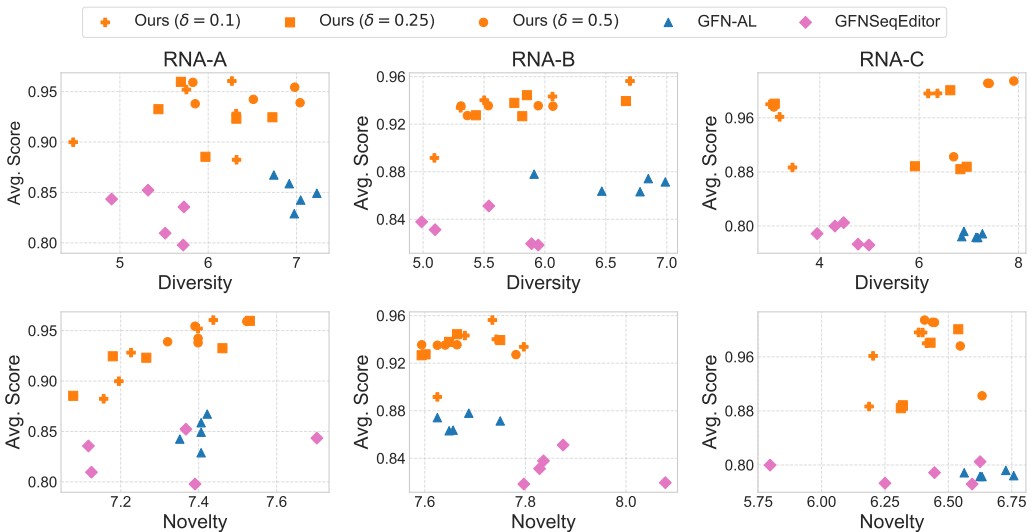

*Figure 13.* Average score and diversity/novelty with on RNA designs with various $\delta$.

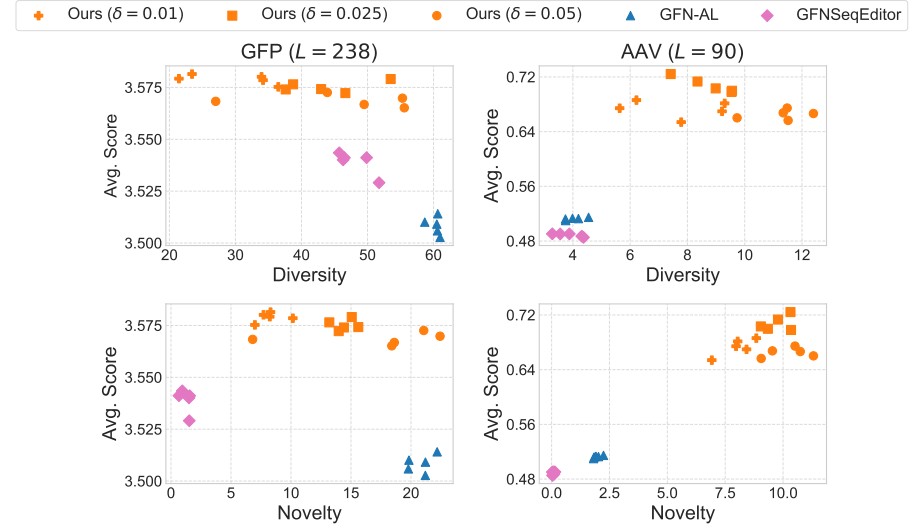

*Figure 14.* Average score and diversity/novelty on protein designs with various $\delta$.

# C. Full results of main results

## C.1. Full results of RNA sequence design

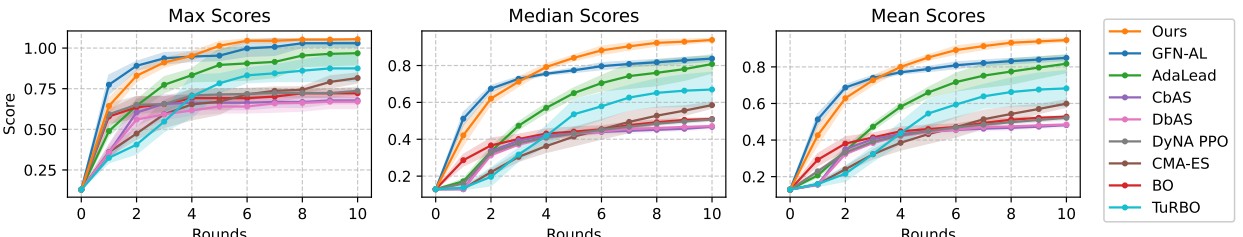

*Figure 15.* The max, median, and mean curve over rounds in RNA-A

*Table 5.* The results of RNA-A after ten rounds.

|  | Max | Median | Mean | Diversity | Novelty |
|---|---|---|---|---|---|
| AdaLead | $0.968 \pm 0.070$ | $0.808 \pm 0.049$ | $0.817 \pm 0.048$ | $3.518 \pm 0.446$ | $6.888 \pm 0.426$ |
| BO | $0.722 \pm 0.025$ | $0.510 \pm 0.008$ | $0.528 \pm 0.004$ | $\mathbf{9.531 \pm 0.062}$ | $5.842 \pm 0.083$ |
| TuRBO | $0.875 \pm 0.078$ | $0.670 \pm 0.093$ | $0.682 \pm 0.096$ | $3.695 \pm 0.166$ | $6.464 \pm 0.759$ |
| CMA-ES | $0.816 \pm 0.030$ | $0.585 \pm 0.016$ | $0.599 \pm 0.020$ | $5.747 \pm 0.110$ | $6.373 \pm 0.159$ |
| CbAS | $0.678 \pm 0.020$ | $0.467 \pm 0.009$ | $0.481 \pm 0.008$ | $9.457 \pm 0.189$ | $5.428 \pm 0.078$ |
| DbAS | $0.670 \pm 0.041$ | $0.472 \pm 0.016$ | $0.485 \pm 0.015$ | $9.483 \pm 0.100$ | $5.450 \pm 0.132$ |
| DyNA PPO | $0.737 \pm 0.022$ | $0.507 \pm 0.007$ | $0.521 \pm 0.009$ | $8.889 \pm 0.034$ | $5.828 \pm 0.095$ |
| GFN-AL | $1.030 \pm 0.024$ | $0.838 \pm 0.013$ | $0.849 \pm 0.013$ | $6.983 \pm 0.159$ | $7.398 \pm 0.024$ |
| GFN-AL + $\delta$-CS | $\mathbf{1.055 \pm 0.000}$ | $\mathbf{0.939 \pm 0.008}$ | $\mathbf{0.947 \pm 0.009}$ | $6.442 \pm 0.525$ | $\mathbf{7.406 \pm 0.066}$ |

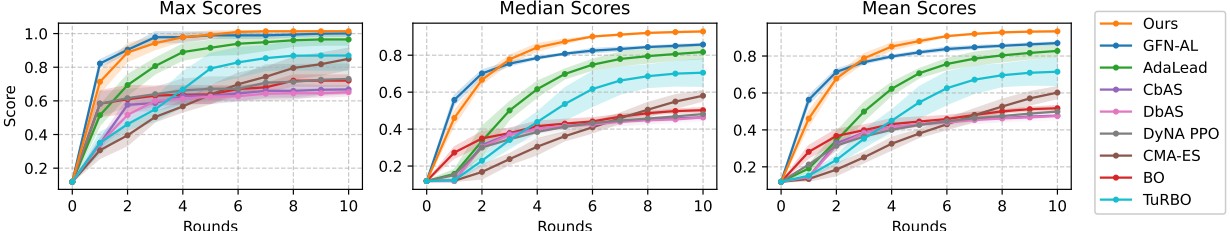

*Figure 16.* The max, median, and mean curve over rounds in RNA-B

*Table 6.* The results of RNA-B after ten rounds.

|  | Max | Median | Mean | Diversity | Novelty |
|---|---|---|---|---|---|
| AdaLead | $0.965 \pm 0.033$ | $0.817 \pm 0.036$ | $0.828 \pm 0.032$ | $3.334 \pm 0.423$ | $7.441 \pm 0.135$ |
| BO | $0.720 \pm 0.032$ | $0.502 \pm 0.013$ | $0.517 \pm 0.014$ | $\mathbf{9.495 \pm 0.103}$ | $5.903 \pm 0.116$ |
| TuRBO | $0.869 \pm 0.086$ | $0.705 \pm 0.073$ | $0.715 \pm 0.073$ | $3.638 \pm 0.173$ | $6.713 \pm 0.560$ |
| CMA-ES | $0.850 \pm 0.063$ | $0.581 \pm 0.028$ | $0.602 \pm 0.032$ | $5.568 \pm 0.365$ | $6.480 \pm 0.200$ |
| CbAS | $0.668 \pm 0.021$ | $0.465 \pm 0.005$ | $0.477 \pm 0.004$ | $9.234 \pm 0.356$ | $5.523 \pm 0.083$ |
| DbAS | $0.652 \pm 0.021$ | $0.463 \pm 0.019$ | $0.475 \pm 0.019$ | $9.019 \pm 0.648$ | $5.537 \pm 0.150$ |
| DyNA PPO | $0.730 \pm 0.088$ | $0.481 \pm 0.028$ | $0.499 \pm 0.029$ | $8.978 \pm 0.196$ | $5.839 \pm 0.198$ |
| GFN-AL | $1.001 \pm 0.016$ | $0.858 \pm 0.004$ | $0.870 \pm 0.006$ | $6.599 \pm 0.384$ | $\mathbf{7.673 \pm 0.043}$ |
| GFN-AL + $\delta$-CS | $\mathbf{1.014 \pm 0.001}$ | $\mathbf{0.929 \pm 0.004}$ | $\mathbf{0.934 \pm 0.003}$ | $5.644 \pm 0.307$ | $7.661 \pm 0.064$ |

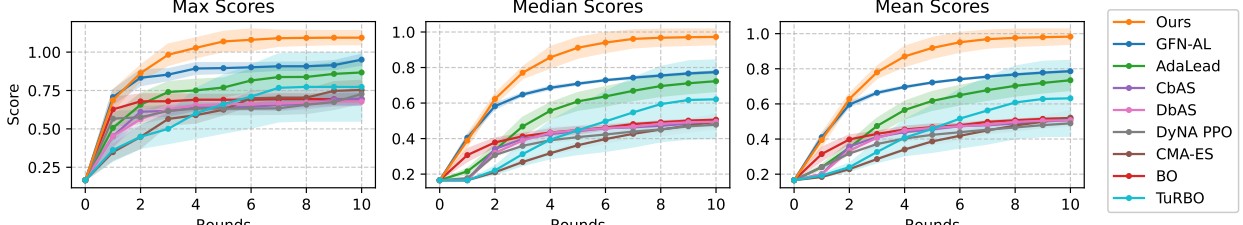

*Figure 17.* The max, median, and mean curve over rounds in RNA-C

*Table 7.* The results of RNA-C after ten rounds.

|  | Max | Median | Mean | Diversity | Novelty |
|---|---|---|---|---|---|
| AdaLead | $0.867 \pm 0.081$ | $0.723 \pm 0.057$ | $0.735 \pm 0.057$ | $3.893 \pm 0.444$ | $5.856 \pm 0.515$ |
| BO | $0.694 \pm 0.034$ | $0.506 \pm 0.003$ | $0.519 \pm 0.003$ | $9.714 \pm 0.054$ | $5.430 \pm 0.043$ |
| TuRBO | $0.773 \pm 0.222$ | $0.621 \pm 0.221$ | $0.632 \pm 0.215$ | $4.403 \pm 1.542$ | $4.906 \pm 1.701$ |
| CMA-ES | $0.753 \pm 0.062$ | $0.496 \pm 0.041$ | $0.521 \pm 0.037$ | $5.581 \pm 0.399$ | $5.019 \pm 0.294$ |
| CbAS | $0.696 \pm 0.041$ | $0.492 \pm 0.018$ | $0.507 \pm 0.017$ | $\mathbf{9.518 \pm 0.310}$ | $5.033 \pm 0.086$ |
| DbAS | $0.678 \pm 0.025$ | $0.495 \pm 0.010$ | $0.508 \pm 0.011$ | $9.249 \pm 0.414$ | $5.128 \pm 0.153$ |
| DyNA PPO | $0.728 \pm 0.060$ | $0.478 \pm 0.015$ | $0.489 \pm 0.015$ | $9.246 \pm 0.086$ | $5.306 \pm 0.124$ |
| GNF-AL | $0.951 \pm 0.034$ | $0.774 \pm 0.004$ | $0.786 \pm 0.004$ | $7.072 \pm 0.163$ | $\mathbf{6.661 \pm 0.071}$ |
| GFN-AL + $\delta$-CS | $\mathbf{1.094 \pm 0.045}$ | $\mathbf{0.972 \pm 0.043}$ | $\mathbf{0.983 \pm 0.043}$ | $6.493 \pm 1.751$ | $6.494 \pm 0.084$ |

## C.2. Full results of TF-Bind-8

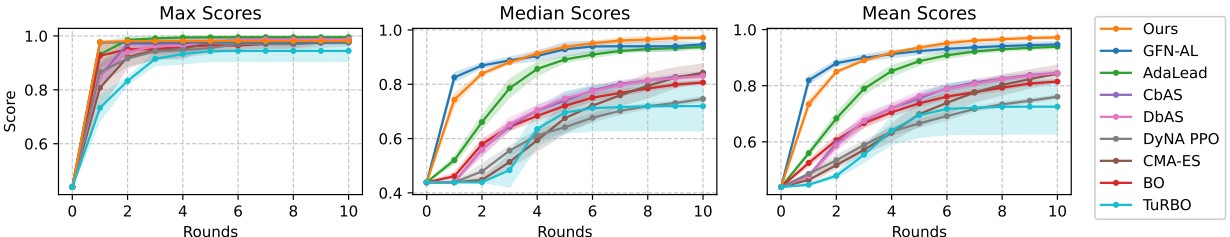

*Figure 18.* The max, median, and mean curve over rounds in TF-Bind-8

*Table 8.* The results of TF-Bind-8 after ten rounds.

|  | Max | Median | Mean | Diversity | Novelty |
|---|---|---|---|---|---|
| AdaLead | **0.995 ± 0.004** | 0.937 ± 0.008 | 0.939 ± 0.007 | 3.506 ± 0.267 | 1.194 ± 0.035 |
| BO | 0.977 ± 0.008 | 0.806 ± 0.007 | 0.815 ± 0.005 | 4.824 ± 0.074 | 1.144 ± 0.029 |
| TuRBO | 0.944 ± 0.039 | 0.719 ± 0.091 | 0.725 ± 0.097 | 4.276 ± 0.680 | 1.005 ± 0.182 |
| CMA-ES | 0.986 ± 0.008 | 0.843 ± 0.032 | 0.843 ± 0.030 | 3.617 ± 0.321 | 1.130 ± 0.083 |
| CbAS | 0.988 ± 0.004 | 0.835 ± 0.011 | 0.845 ± 0.009 | 4.662 ± 0.079 | 1.134 ± 0.021 |
| DbAS | 0.987 ± 0.004 | 0.831 ± 0.005 | 0.845 ± 0.005 | 4.694 ± 0.056 | 1.141 ± 0.047 |
| DyNA PPO | 0.977 ± 0.013 | 0.746 ± 0.010 | 0.761 ± 0.006 | 4.430 ± 0.030 | 1.120 ± 0.021 |
| GFN-AL | 0.976 ± 0.002 | 0.947 ± 0.004 | 0.947 ± 0.009 | 3.158 ± 0.166 | **2.409 ± 0.071** |
| GFN-AL + $\delta$-CS | 0.981 ± 0.002 | **0.971 ± 0.006** | **0.972 ± 0.005** | **1.277 ± 0.182** | 2.237 ± 0.356 |

## C.3. Full results of protein design

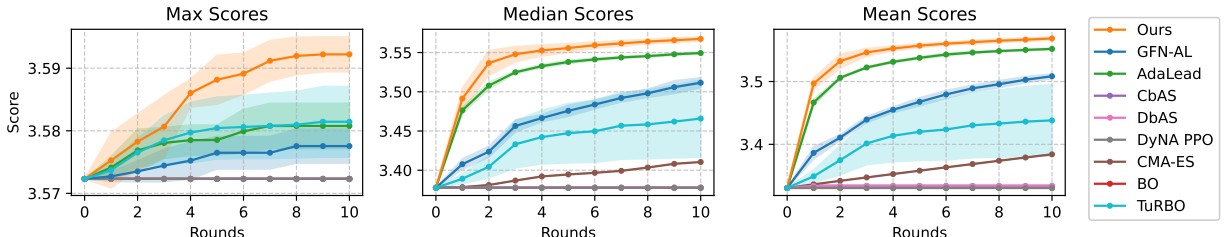

*Figure 19.* The max, median, and mean curve over rounds in GFP

*Table 9.* The results of GFP after ten rounds.

|  | Max | Median | Mean | Diversity | Novelty |
|---|---|---|---|---|---|
| AdaLead | $3.581 \pm 0.004$ | $3.549 \pm 0.002$ | $3.552 \pm 0.002$ | $47.237 \pm 1.213$ | $1.467 \pm 0.094$ |
| BO | $3.572 \pm 0.000$ | $3.378 \pm 0.000$ | $3.331 \pm 0.000$ | $\mathbf{62.955 \pm 0.000}$ | $0.000 \pm 0.000$ |
| TuRBO | $3.581 \pm 0.006$ | $3.466 \pm 0.049$ | $3.438 \pm 0.057$ | $51.112 \pm 6.802$ | $0.584 \pm 0.379$ |
| CMA-ES | $3.572 \pm 0.000$ | $3.410 \pm 0.000$ | $3.384 \pm 0.000$ | $58.299 \pm 0.000$ | $0.000 \pm 0.000$ |
| CbAS | $3.572 \pm 0.000$ | $3.378 \pm 0.000$ | $3.334 \pm 0.002$ | $62.926 \pm 0.139$ | $0.009 \pm 0.012$ |
| DbAS | $3.572 \pm 0.000$ | $3.378 \pm 0.000$ | $3.334 \pm 0.002$ | $62.926 \pm 0.139$ | $0.009 \pm 0.012$ |
| DyNA PPO | $3.572 \pm 0.000$ | $3.378 \pm 0.000$ | $3.331 \pm 0.000$ | $\mathbf{62.955 \pm 0.000}$ | $0.000 \pm 0.000$ |
| GFN-AL | $3.578 \pm 0.003$ | $3.511 \pm 0.006$ | $3.508 \pm 0.004$ | $60.278 \pm 0.819$ | $\mathbf{20.837 \pm 0.916}$ |
| GFN-AL + $\delta$-CS | $\mathbf{3.592 \pm 0.003}$ | $3.567 \pm 0.003$ | $\mathbf{3.569 \pm 0.003}$ | $46.255 \pm 10.534$ | $17.459 \pm 5.538$ |

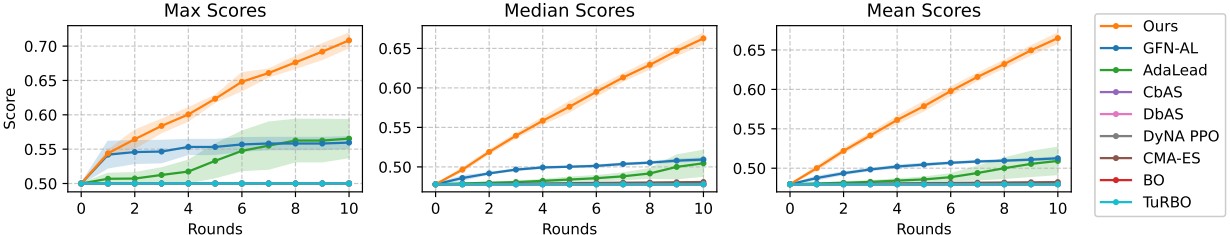

*Figure 20.* The max, median, and mean curve over rounds in AAV

*Table 10.* The results of AAV after ten rounds.

|  | Max | Median | Mean | Diversity | Novelty |
|---|---|---|---|---|---|
| AdaLead | $0.565 \pm 0.027$ | $0.505 \pm 0.016$ | $0.509 \pm 0.017$ | $5.693 \pm 0.946$ | $2.133 \pm 1.266$ |
| BO | $0.500 \pm 0.000$ | $0.478 \pm 0.000$ | $0.480 \pm 0.000$ | $4.536 \pm 0.000$ | $0.000 \pm 0.000$ |
| TuRBO | $0.500 \pm 0.000$ | $0.478 \pm 0.000$ | $0.480 \pm 0.000$ | $4.533 \pm 0.006$ | $0.002 \pm 0.003$ |
| CMA-ES | $0.500 \pm 0.000$ | $0.481 \pm 0.000$ | $0.482 \pm 0.000$ | $4.148 \pm 0.000$ | $0.000 \pm 0.000$ |
| CbAS | $0.500 \pm 0.000$ | $0.478 \pm 0.000$ | $0.480 \pm 0.000$ | $4.545 \pm 0.018$ | $0.002 \pm 0.003$ |
| DbAS | $0.500 \pm 0.000$ | $0.478 \pm 0.000$ | $0.480 \pm 0.000$ | $4.545 \pm 0.018$ | $0.002 \pm 0.003$ |
| DyNA PPO | $0.500 \pm 0.000$ | $0.478 \pm 0.000$ | $0.480 \pm 0.000$ | $4.536 \pm 0.000$ | $0.000 \pm 0.000$ |
| GFN-AL | $0.560 \pm 0.008$ | $0.509 \pm 0.002$ | $0.513 \pm 0.002$ | $4.044 \pm 0.303$ | $1.966 \pm 0.157$ |
| GFN-AL + $\delta$-CS | $\mathbf{0.708 \pm 0.010}$ | $\mathbf{0.663 \pm 0.007}$ | $\mathbf{0.665 \pm 0.006}$ | $\mathbf{11.296 \pm 0.865}$ | $\mathbf{10.233 \pm 0.822}$ |

