# OpenReview forum: "Improved Off-policy Reinforcement Learning in Biological Sequence Design"
_ICML.cc/2025/Conference — ICML 2025 poster_

### Official Review · Reviewer_GMEp · 2025-03-12

**Overall Recommendation:** 3

**Summary:**

This work focuses on the problem of biological sequence design, recognizing that trained proxy scoring models often produce unreliable predictions. To address this challenge, they introduce restrictions into GFlowNet exploration by implementing a masking mechanism that increases the probability of exploring reliable data points. Specifically, adding masks to certain tokens in the offline dataset allows for exploration at those positions, while unmasked positions directly utilize offline data through teacher forcing. Through this approach, the authors' proposed δ-CS method supports more reliable exploration, achieving excellent results across multiple datasets (DNA, RNA, peptide design) and attaining Pareto optimality by balancing diversity and fitness.

## update after rebuttal
Thank the authors for their rebuttal, which addressed many of my concerns, especially by adding experiments on incorporating $\delta$-CS results with additional off-policy methods. As a result, I have increased my score from 2 to 3. However, for a primarily empirical work like this, I still believe that the current depth of investigation into the effects of $\delta$-CS is insufficient.

**Claims And Evidence:**

See Other Strengths And Weaknesses and Questions For Authors.

**Essential References Not Discussed:**

See Other Strengths And Weaknesses and Questions For Authors.

**Experimental Designs Or Analyses:**

See Other Strengths And Weaknesses and Questions For Authors.

**Methods And Evaluation Criteria:**

See Other Strengths And Weaknesses and Questions For Authors.

**Other Comments Or Suggestions:**

See Other Strengths And Weaknesses and Questions For Authors.

**Other Strengths And Weaknesses:**

## Strengths:
1. The paper is well-written, provides a detailed literature review, and clearly explains both the motivation and methods.
2. Experimental details are provided thoroughly, and source code is also provided, ensuring high reproducibility.
3. The motivation makes sense, as considering the unreliability of proxies in biological sequence optimization is indeed an important problem.

## Weaknesses:
1. While the masking approach proposed here is conceptually novel, I find the term "mask" potentially misleading. In sequence modeling machine learning, mask usage typically refers to approaches like BERT, where certain tokens are masked within a sequence while content on both sides remains visible, enabling bidirectional modeling. After reading the abstract, I initially expected the work to employ this type of modeling. Upon further reading, it appears that the "mask" here merely serves as a "selective exploration" training method. Specifically, in GPT's next token prediction, training traditionally requires teacher forcing for every token through trajectories from dataset $D$. The masking here simply selects certain tokens that are allowed free exploration, while maintaining an autoregressive training approach at its core. This usage of "mask" may confuse readers. I would like the authors to clarify their rationale for using the "mask" concept within an autoregressive training framework. Without sufficient justification, I recommend the authors consider alternative terminology.

2. The rationale for using GFlowNet remains unclear. As mentioned in Weakness 1, the essence of the method is conducting selective exploration on offline data to discover new trajectories, but this approach doesn't seem strongly coupled with GFlowNets. I believe the authors should include other methods capable of learning from offline data in their baselines, and then demonstrate the superiority of $\delta$-CS as a plug-and-play component, proving its advantages over alternative approaches.

3. The authors' motivation is premised on proxy models producing severe unreliable predictions (Figure 2), leading them to reduce exploration of unreliable data points. However, I note that the proxy used is very simple (a CNN with no additional regularization). The authors should further discuss whether more advanced proxy training approaches could mitigate this issue (which represents an alternative solution to unreliability, as in [2]).

4. The DNA design experiments (TFBS binding sites <10bp) are overly simplistic (as mentioned on line 261). The authors should consider more complex experiments involving Cis-regulatory Element design as demonstrated in recent works ([1][2][3][4]).

5. [1] shares a similar approach with this work, using prediction uncertainty to adjust the optimization process. The difference is that [1] incorporates uncertainty into the reward function, while the authors use it to restrict exploration probability from offline data. I would like the authors to discuss the differences between these two approaches. Additionally, [1] employs different uncertainty estimation methods, and I would like the authors to discuss how their uncertainty estimation approach differs from that in [1].

## References

[1] Uehara, Masatoshi, et al. "Bridging model-based optimization and generative modeling via conservative fine-tuning of diffusion models." Advances in Neural Information Processing Systems 37 (2024): 127511-127535.

[2] Reddy, Aniketh Janardhan, et al. "Designing Cell-Type-Specific Promoter Sequences Using Conservative Model-Based Optimization." Advances in Neural Information Processing Systems 37 (2024): 93033-93059.

[3] Yang, Zhao, et al. "Regulatory DNA Sequence Design with Reinforcement Learning." The Thirteenth International Conference on Learning Representations.

[4] Wang, Chenyu, et al. "Fine-Tuning Discrete Diffusion Models via Reward Optimization with Applications to DNA and Protein Design." The Thirteenth International Conference on Learning Representations.

**Questions For Authors:**

1. Do we actually know the distribution $P_{D_{t-1}}$? I understand the authors may intend to construct a discrete distribution using the weight in line 132, but this requires a more rigorous definition of the distribution.

2. What is the significance of introducing $\tilde{x}$ on line 160? This notation is not used in the formula, which only requires $\tilde{e}_t$. This aligns with my concern in Weakness 1, and actually the modeling here doesn't require a complete masked $\tilde{x}$, so is using "mask" as terminology for the method appropriate?

3. Line 22 mentions that the biggest drawback of DyNA PPO is its inability to utilize offline data, but I believe this might be solvable. Couldn't DyNA PPO directly leverage offline data through teacher-forcing methods?

**Relation To Broader Scientific Literature:**

See Other Strengths And Weaknesses and Questions For Authors.

**Theoretical Claims:**

N/A

---

> ### Author Rebuttal · Authors · 2025-03-31
>
> >W1, Q2: The term "mask" is confusing
>
> Because our infilling sites are uniformly distributed across the entire sequence, we refer to them as “masks,” similar to BERT. However, we recognize that this term can be confusing. Alternatively, we could describe the algorithm as random teacher forcing: some tokens are predicted while others remain identical to the original sequence. This approach also removes the need for $\tilde{x}$ in the equation, since it is not directly used as input by the neural network.
>
> > W2: 𝛿CS with other algorithms beyond GFlowNet
>
> 𝛿-CS can be applied to any off-policy RL method; we chose GFlowNets because they are representative method in this domain. To demonstrate $\delta$-CS as a plug-and-play component, we applied 𝛿-CS to Soft Q-Learning (SQL), which is a representative off-policy RL method in the general domain:
>
>
> | Hard TFBind8| t=1 | t=5 | t=10 |
> | - | - | - | - |
> | SQL | 0.531 (0.011) | 0.879 (0.014) | 0.936 (0.008) |
> | SQL + 𝛿-CS (𝛿=0.5) |0.546 (0.013) | 0.975 (0.011) | 0.993 (0.007) |
>
> ---
>
> > W3: Conservative proxy
>
> We acknowledge conservative model-based optimization (COMs) methods, which aim to construct robust proxy models [1, 2]. However, mere conservativeness is insufficient for active learning scenarios, where active querying for information gain from the oracle is essential. Although COMs perform well in offline model-based optimization (MBO), our results show they underperform in active learning settings. Specifically, COMs that impose conservativeness through adversarial smoothness in proxy training (Trabucco et al., 2021) achieve poorer outcomes than alternative methods.
>
> Our method shares the philosophy that conservatism is beneficial but differs significantly in implementation. Instead of embedding conservativeness directly into proxy training, we integrate it into the search process itself. This approach preserves high information gain through explicit uncertainty modeling while ensuring conservatively robust optimization, effectively balancing uncertainty-driven exploration with robustness. We will put this discussion at the main paper. Thanks.
>
> Trabucco et al. "Conservative objective models for effective offline model-based optimization." ICML, 2021.
>
> ---
> >W4: Validation in complex biological sequence task
>
> We have already included longer sequence benchmarks (such as GFN and AAV), which are widely recognized and commonly used in the literature. While TFB tasks are indeed simpler and considered toy benchmarks, they remain popular for initial evaluations. We agree that incorporating additional real-world DNA benchmarks could further strengthen our evaluation.
>
> Regarding the suggested benchmarks, we noticed that the majority of existing literature [1,2,4] focuses on diffusion-based algorithms, which fall outside the scope of our research (therefore we focused to benchamrk [3]). Although we aimed to replicate results from TACO [3], which employs autoregressive models, their dataset and pre-trained checkpoints are unfortunately not publicly available. Consequently, extending our method to include these benchmarks within the rebuttal period is challenging. However, we commit to evaluating our method on longer DNA benchmarks by the camera-ready deadline, and we anticipate similar performance trends to those observed in GFP and AAV tasks.
>
> ---
>
> >W5: $\delta$-CS vs. conservative reward
>
> As discussed in W3, our approach still can utilize acquisition function like UCB while introducing conservatism to avoiding overoptimization. Though we have different approach, we can utilize the uncertainty oracle as proposed in [1] since we do not any restriction for the way of measuring uncertainty; this will be an interesting future work.
>
> We'll include this discussion into our revised manuscript
>
> ---
> > Q1. Do we actually know the distribution $P_{D_{t-1}}$?
>
> We define a discrete distribution by normalizing the weights in line 132. Specifically, $P_{D_{t-1}}(x) \propto w(x; D_{t-1}, k),$ where $k$ is a hyperparameter controlling the distribution’s peakiness.
>
> ---
> > Q3. Couldn't DyNA PPO directly leverage offline data through teacher-forcing methods?
>
> We agree that teacher-forcing offers a way to incorporate offline data into DyNA-PPO. However, because PPO is inherently on-policy, directly leveraging offline data remains challenging without methods like importance sampling, which can introduce high variance. While teacher-forcing also can be used to pretrain the policy pretraining from offline sequences, it may lead to overfitting to the offline distribution since the reward information is not used during pretraining. To further verify this point, we conducted DyNA PPO experiments where we first pretrain the policy via teacher-forcing on offline data and then fine-tune it with PPO on TFBind8:
>
> ||Top-128 Mean|
> |--|--|
> |Dyna PPO|0.761 ± 0.006|
> |Dyna PPO + teacher - forcing pretraining|0.777 ± 0.001 |
> |GFN-AL | 0.947 ± 0.009 |
> |GFN-AL + ours | 0.972 ± 0.005 |

---

> > ### Comment · Reviewer_GMEp · 2025-04-02
> >
> > I thank the authors for their comprehensive rebuttal, which has effectively addressed many of my previous concerns. However, I still have several points that warrant further discussion:
> >
> > 1. While the proposed method is positioned as a "plug-and-play" enhancement for off-policy methods, the current experimental validation remains somewhat limited. Despite the additional experiments provided in the rebuttal, only one extra method was evaluated, and this evaluation was conducted solely on the relatively less complex TF Binding Task.
> >
> > 2. Could you elaborate on the specific conditions under which the active learning setting offers advantages over the offline MBO setting for biological sequence design problems? Please provide further justification for your specific focus on the active learning paradigm in this work.

---

> > > ### Author Response · Authors · 2025-04-06
> > >
> > > **Thank you for the additional feedback. We are glad to hear that most of your concerns have been addressed. Below, we respond to the remaining points.**
> > >
> > > ----
> > >
> > > > 1. While the proposed method is positioned as a "plug-and-play" enhancement for off-policy methods, the current experimental validation remains somewhat limited. Despite the additional experiments provided in the rebuttal, only one extra method was evaluated, and this evaluation was conducted solely on the relatively less complex TF Binding Task.
> > >
> > >
> > > **Mean Top-128**
> > > |  | GFP | AAV |
> > > | - | - | - |
> > > | SQL | 3.331 (0.0) | 0.480 (0.0) |
> > > | SQL + 𝛿-CS | 3.573 (0.003) | 0.495 (0.001) |
> > > | VarGrad | 3.331 (0.0) | 0.480 (0.0) |
> > > | VarGrad + 𝛿-CS | 3.567 (0.003) | 0.668 (0.011) |
> > > | Soft PCL ($\lambda$=0.9) | 3.331 (0.0) |  0.480 (0.0)|
> > > | Soft PCL ($\lambda$=0.9) + 𝛿-CS | 3.578 (0.004) | 0.622 (0.011) |
> > >
> > >
> > > As you noted, we expanded our SQL experiments to the main protein tasks (GFP and AAV) and added two additional soft off-policy RL algorithms, Log-Partition Variance Gradient (VarGrad) [1] and Soft Path Consistency Learning (PCL) [2,3,4], resulting in four baselines (TB, SQL, VarGrad, Soft PCL). **Our results show that 𝛿-CS improves performance across all of them, highlighting its plug-and-play versatility.**
> > >
> > > **Discussion on off-policy RL algorithms:** Briefly, TB and VarGrad apply constraints on entire trajectories, leading to higher variance but lower bias. TB explicitly estimates flows/values (partition function of terminal state), while VarGrad does so implicitly through batch averaging. Soft PCL operates on sub-trajectories (similar to a TD-λ approach), striking a variance-bias balance via trajectory length. SQL uses one-step transitions, lowering variance but increasing bias. In these protein design tasks, where global sequence properties matter, full-trajectory methods (TB, VarGrad) generally seem to outperform sub-trajectory or one-step approaches.
> > >
> > > For further analysis connecting these off-policy RL methods to discrete sampling (including biological design) see [5]. We will include these results and related discussion in the main paper. Thank you for the valuable feedback.
> > >
> > >
> > > > 2. Could you elaborate on the specific conditions under which the active learning setting offers advantages over the offline MBO setting for biological sequence design problems? Please provide further justification for your specific focus on the active learning paradigm in this work.
> > >
> > > Active learning naturally matches how biological sequence design often proceeds: researchers propose candidate sequences, test them in assays or models, and use those results to guide subsequent rounds of design. This iterative cycle quickly pinpoints promising variants by balancing exploration (searching new sequence space) and exploitation (optimizing designs in safe region). In practice, multiple verification steps—such as in vitro (cell-level experiments) assays or in vivo (animal or human studies)—can serve as oracle queries, feeding fresh data back into the surrogate model [6]. Meanwhile, offline MBO relies on a single batch of pre-collected data and is useful when further experimentation is infeasible (e.g., out-of-budget, safety constraints) or during final decision stages after active learning.
> > >
> > > ----
> > >
> > >
> > > [1] Lorenz et al., "Vargrad: a low-variance gradient estimator for variational inference.", NeurIPS 2020
> > >
> > > [2] Nachum et aL., "Bridging the Gap Between Value and Policy Based Reinforcement Learning.", NeurIPS 2017
> > >
> > > [3] Chow et al., "Path consistency learning in tsallis entropy regularized mdps", NeurIPS 2018
> > >
> > > [4] Madan et al., "Learning GFlowNets from partial episodes for improved convergence and stability", ICML 2023
> > >
> > > [5] Deleu et al., "Discrete Probabilistic Inference as Control in Multi-path Environments", UAI 2024
> > >
> > > [6] Jain, Moksh, et al. "Biological sequence design with gflownets." International Conference on Machine Learning. PMLR, 2022.

---

### Official Review · Reviewer_GtXk · 2025-03-12

**Overall Recommendation:** 3

**Summary:**

The manuscript proposes a conservative search method applied in GFlowNets for Biological Sequence Design. The proposed method restricts the number of mutations based on the length of the sequence and the prediction uncertainty. Experiments show that this sampling methodology stabilizes the training of GFlowNets, improving performance over GFlowNet-AL and other traditional baselines. These results highlight that simple methodologies for restricting the search space are effective when the functionality landscape is usually located around the initial sequence of interest.

## update after rebuttal

The authors have addressed many of my concerns. I updated my score accordingly.

**Claims And Evidence:**

The claims made by the manuscript are supported by clear evidence.

**Essential References Not Discussed:**

Additional references related to the application of RL to Protein Engineering like [REF1] and [REF2] seem to be missing.

[REF1] Wang, Yi, et al. "Self-play reinforcement learning guides protein engineering." Nature Machine Intelligence 5.8 (2023): 845-860.
[REF2] Lee, Minji, et al. "Robust optimization in protein fitness landscapes using reinforcement learning in latent space." ICML (2024).

**Experimental Designs Or Analyses:**

The reviewer has clarification questions regarding the active learning setting and the oracles used for evaluation. Specifically,

1. How was the active learning setting set for the proposed method and the baseline methods?
2. How is the oracle performing for sequences that are far in terms of mutations from the wild type? The values for the GFP performance graph seem high even for methods that have problems generating long sequences like DynaPPO.

**Methods And Evaluation Criteria:**

The evaluation criteria make sense for the problem investigated.

**Other Comments Or Suggestions:**

1. (Related Work) GFlowNets are introduced on Page 5 after there are other follow-up works to GFlowNets introduced in Page 4. The Related Work section should be re-organized.
2. The sentence in line 212: “capitalizing on both the high novelty offered by GFlowNets” is counterintuitive as your method proposes restricting the novelty of the GFlowNet generation given its tendency to generate out-of-distribution samples.

**Other Strengths And Weaknesses:**

Strengths:
1. The manuscript adapts GFlowNets by proposing a conservative search framework, i.e., sampling a limited number of mutations from initial state sequences, to address the issue of generating out-of-distribution samples when handling large-length sequences.
2. The idea of using restricted exploration to improve the training efficiency of GFlowNets is an interesting direction.
3. The proposed adaptive conservativeness approach is adjusted for each data point based on prediction uncertainty.

Weaknesses:
1. The main concern of the reviewer is the reliability of the results presented in the manuscript. The oracle and active learning settings are not well defined. Additionally, there are some concerns regarding dataset splits used for some of the tasks presented.
2. As mentioned by the authors as one of the main limitations, the restricted exploration strategy does not solve different drawbacks from active learning. Restricting the number of mutations is likely to improve the results for landscapes in which most of the functional sequences are close to the main wild-type sequence.
3. Even though the title suggests a general method analysis for different RL algorithms, the experiments are only extending a GFlowNet as the main policy.

**Questions For Authors:**

1. What is the meaning of “large-scale” settings in line 36? From my understanding, it means that GFlowNets achieve poor performance for long sequences? The authors should clarify this definition.
2. (Title) The title suggests that the proposed conservative search can be applied to other RL-based algorithms but only test to GFlowNet. For other algorithms, the proposed policy is only applied for a setting with discrete actions and with a sampling order that gives enough context to propose mutations to the masked positions? If this is the case, the reviewer wonders about the advantage or improvement in performance over sampling mutations using a masked protein language model, for example.
3. The methodology for generating the datasets seems critical. How were the initial sequences for GFP generated? With random mutations from the wild-type? Given the functional landscape of GFP this might lead to data leakage. A split like the one proposed by [REF1] might be needed.
[REF1] Kirjner, Andrew, et al. "Improving protein optimization with smoothed fitness landscapes." ICLR (2023).
4. All baselines achieve very high performance on the GFP dataset. Given that sequences with more than 10 mutations are very unlikely to be functional, the reviewer suggests showing the number of mutations from the wild-type for each of the methods. A metric showing the distance from the wild-type for the top-scoring mutants might be needed, as shown in [REF2]. Mutants far from the wild type with high predicted functionality might suggest a problem with the oracle's reliability.
[REF2] Lee, Minji, et al. "Robust optimization in protein fitness landscapes using reinforcement learning in latent space." ICML (2024).
5.  It is not clear how the active learning setting is defined for the proposed method and baselines. Additionally, it is also unclear if the oracle is used only for evaluation for all the methods.
6. Given that the positions to mutate are pre-sampled, it would be interesting to have baselines mutating these same positions randomly or using a masked protein language model to compare the performance.
7. The manuscript needs a better explanation and intuition of how fixing positions changes the training stability of GFlowNets.

**Relation To Broader Scientific Literature:**

The key contributions of the paper related to the broader scientific literature are:

1. Proposing a conservative search approach that can be combined to solve sampling issues in the training of GFlowNets.
2. It applies an adaptive conservativeness approach that is dependent on the oracle prediction uncertainty.

**Theoretical Claims:**

The mathematical of the proposed method seems correct.

---

> ### Author Rebuttal · Authors · 2025-03-31
>
> ### Active learning setting
>
> We follow the existing generative active learning settings (GFN-AL [1], FLEXS [2]), which are standard in this field. As noted in our main text and appendix, we perform active learning as follows:
>
> Starting with an initial dataset $D_0$, each active learning round t consists of three steps:
> - (Step A) train the proxy model on the current dataset $D_{t-1}$;
> - (Step B) train the generative policy using the proxy and proposing a batch of B new sequences;
> - (Step C) evaluate those sequences using Oracle and add them to the dataset.
>
> We run 10 active learning rounds with a batch size of 128. We have the same limited oracle calls for all methods. At each round, we assess performance using the top-K sequences (with K=128).
> We adopt the initial dataset of DNA and RNA from [1] and [3], respectively. For GFP and AAV, we generated the initial dataset using wild-type, but as discussed in the following questions, we have re-conducted experiments using the dataset from [REF1, 2].
>
> ### Experimental setting for GFP
>
> First of all, our GFP benchmark is quite standard in this field; we use TAPE [4], one of the widely adopted benchmarks in representative studies (e.g., [2], [7]). The benchmarking methods for the GFP task are diverse (e.g., TAPE [4], Design-Bench [5]; see [6] for detailed comparisons among them). Additionally, scoring normalization varies across studies, with some methods employing normalized scores [REF1, 2] and others using unnormalized scores [1, 2, 7].
>
> As suggested, we have conducted experiments by following the experimental setting of LatProtRL [REF2]: we evaluate 256 sequences in 15 active learning rounds by adopting the oracle function with the given initial dataset that is split following [REF1]. As with [REF2], we report the median value of the top-128 sequences over 5 independent runs. The results for AdaLead and LatProtRL are directly obtained from [REF2]. Our approach achieved better performance than LatProtRL and substantially improved GFN-AL.
>
> | | GFP Medium |  GFP Hard |
> | - | - | - |
> | AdaLead | 0.93 (0.0) | 0.75 (0.1) |
> | LatProtRL | 0.93 (0.0) | 0.85 (0.0) |
> | GFN-AL (𝛿=1) | 0.21 (0.0) | 0.09 (0.0) |
> | GFN-AL + 𝛿-CS (𝛿=0.05) | 0.57 (0.0) | 0.60 (0.1) |
> | GFN-AL + 𝛿-CS (𝛿=0.01) | 1.06 (0.1) | 0.86 (0.0) |
>
> In the revised manuscript, we will discuss the challenges in GFP with additional experimental results.
>
> ### Distance to the wild-type and restricted exploration in GFP
>
> We assume the oracle to be perfect, as with previous works. The performance of proxies typically degrades when predicting samples far from the offline data distribution, which is particularly evident in our constrained search.
>
> We agree that incorporating distance from wild-type sequences leads to more realistic benchmarking. However, we focus on proposing a broadly applicable off-policy search method, not benchmarking itself. Importantly, our approach is compatible with distance-based constraints, as they are orthogonal to our method. To demonstrate this, we applied δ-CS to Proximal Exploration (PEX) [7], which explicitly encourages proximity to the wild type—see results below.
>
> | GFP Hard  | 𝛿-CS | 𝛿-CS + PEX |
> |-|-|-|
> | Fitness | 0.86 (0.0) | 1.08 (0.0) |
> | Distances to wt | 21.31 (1.7) | 7.47 (0.6) |
>
> ### Validation with other off-policy RL algorithms
> As suggested, we applied our method to other RL methods: soft Q learning. Please refer to the answer to reviewer 6J1t (W3).
>
> ### Other baselines
> As suggested, we compare with random mutation and LS-GFN. LS-GFN is our special case where the masking positions are fixed as the last 𝛿L tokens.
>
> | | GFP Hard |
> | - | - |
> | 𝛿-CS (𝛿=0.01) | 0.86 (0.0) |
> | Random (𝛿=0.01) | 0.27 (0.0) |
> | LS-GFN (𝛿=0.01)  | 0.34 (0.0) |
>
> The table shows that fixing masking positions can overly restrict the search space. However, specifying more significant positions in sequences based on domain knowledge can be more efficient than randomly choosing the positions.
>
> ### Related works
> As suggested, we'll include the missing references and re-organize Sec 5.
>
> ### Other comments
> > DyNa PPO in GFP
>
> In Fig 4, the mean score of Dyna PPO remains unchanged over active rounds, which means it fails to generate sequences with higher scores than the initial dataset. Note that GFP scores are not normalized.
>
> > "Large-scale"
>
> We use "large-scale" to denote longer sequences, especially with large action space, like protein design tasks.
>
> > line 212
>
> We'll revise it.
>
>
> [1] Jain et al. "Biological sequence design with GFlowNets." ICML (2022)
>
> [2] Sinai et al. "Adalead" (2020)
>
> [3] Kim et al. "Bootstrapped training of score-conditioned generator for offline design of biological sequences." NIPS (2023)
>
> [4] Rao et al. "Evaluating protein transfer learning with TAPE." NIPS (2019)
>
> [5] Trabucco et al. "Design-bench" ICML, 2022.
>
> [6] Surana et al. "Overconfident oracles." (2025).
>
> [7] Ren et al. "Proximal exploration for model-guided protein sequence design." ICML, 2022.

---

> > ### Comment · Reviewer_GtXk · 2025-04-03
> >
> > Thank you for answering my questions.
> >
> > I have increased my score after the rebuttal from the authors.
> >
> > I have the following additional comments/questions:
> >
> > 1. (Initial Dataset D_0) For the GFP and AAV experiments, I still think that the initial dataset should be from low-functional initial sequences, and not only from mutations from the wild type sequence.
> >
> > 2. (Distance to the Wild-Type) For the results presented in the section evaluating the distance from the wild-type, when combining the proposed method with PEX is the wild-type sequence result used during optimization?
> >
> > 3. (Other Baselines) My suggestion was to, for the same positions mutated by the proposed method, try random mutations and mutations performed by a language model. From my understanding, in the rebuttal table, the positions mutated for "Random" and "LS-GFN" are different than the ones for the proposed method?

---

> > > ### Author Response · Authors · 2025-04-06
> > >
> > > **Thanks for the additional feedback, and we appreciate some of your concerns were addressed.**
> > >
> > > > 1. (Initial Dataset D_0) For the GFP and AAV experiments, I still think that the initial dataset should be from low-functional initial sequences, and not only from mutations from the wild type sequence.
> > >
> > > **We want to clarify that we already used the initial dataset consisting of low-functional sequences in the additional experiments**. Specifically, we directly adopt the initial dataset from Lee et al. (2024) which satisfy the condtions the reviewer mentioned. Note that all additional experiments related GFP in the rebuttal use the initial dataset from Lee et al. (2024). We will include these additional experiments in the revised manuscript.
> > >
> > >
> > > > 2. (Distance to the Wild-Type) For the results presented in the section evaluating the distance from the wild-type, when combining the proposed method with PEX is the wild-type sequence result used during optimization?
> > >
> > >
> > > We combined our method with PEX follows these steps each round:
> > > (a) train the policy and propose new sequences,
> > > (b) select 𝐵 sequences to query via PEX by
> > > - evaluating all sequences with the proxy,
> > > - measuring their distances to the wild type, and
> > > - finding proximal frontier using the proxy value and disctance
> > >
> > >
> > >
> > >
> > > > 3. (Other Baselines) My suggestion was to, for the same positions mutated by the proposed method, try random mutations and mutations performed by a language model. From my understanding, in the rebuttal table, the positions mutated for "Random" and "LS-GFN" are different than the ones for the proposed method?
> > >
> > > The "Other Baselines" section is for question 6 and 7; we grouped the corresponding results into a combined section due to space constraints, and we apologize for any confusion.
> > >
> > > > Q6. Given that the positions to mutate are pre-sampled, it would be interesting to have baselines mutating these same positions randomly or using a masked protein language model to compare the performance.
> > >
> > > **(Baseline 1: random mutations)** We already included the random mutation baseline, where mutations are applied to the same masked positions selected by our noise injection policy, to isolate the effect of the denoising policy. This allows us to assess whether performance improvements come from the conservative search mechanism itself or simply from where mutations are applied.
> > >
> > > **(Baseline 2: pLM)** We agree that comparing our method to a masked protein language model (pLM) could offer a useful perspective. However, we deliberately focused on an active learning setting where the model is trained solely on a limited offline dataset, ensuring no information leakage from the test set. In contrast, many pretrained pLMs are trained on large public databases, making it difficult to guarantee that they have not seen target sequences or close variants, potentially giving them an unfair advantage.
> > > We acknowledge the value of pretrained models when properly controlled. In future work, one could retrain a masked pLM from scratch on a dataset that is disjoint from the test set to enable a fairer and more rigorous comparison with active learning approaches.
> > >
> > >
> > > > Q7. The manuscript needs a better explanation and intuition of how fixing positions changes the training stability of GFlowNets.
> > >
> > > The experiments with LS-GFN are included to give an intuition of how fixing positions affects GFlowNet training (Q7). LS-GFN performs a back-and-forth search by fixing the masked positions to the last 𝛿𝐿 tokens (a special case of our framework). As the results show, fixing positions in this way can overly restrict the search space, leading to suboptimal exploration and reduced diversity.
> > >
> > > That said, if the fixed positions are chosen based on domain knowledge (e.g., known functional regions), this constraint could be beneficial by focusing exploration on more meaningful parts of the sequence. In contrast, randomly selecting positions, used in our main method, offers broader and more stable exploration when such prior knowledge is unavailable.

---

### Official Review · Reviewer_JojN · 2025-03-15

**Overall Recommendation:** 3

**Summary:**

The paper proposes δ-CS, a novel off-policy RL  approach for biological sequence design. It addresses the challenge of proxy misspecification, where proxy models used for sequence evaluation are unreliable on ood inputs. The method is integrated into GFlowNets, and works by injecting and denoising noise into high-score sequences with dynamically adjusted conservativeness. Experiments show that δ-CS significantly improves GFlowNets by balancing sequence exploration and robustness, leading to improved DNA, RNA, protein, and peptide design outcomes.

**Claims And Evidence:**

Yes. Fig3 and table 1 supports the claim that δ-CS improves robustness by restricting policy exploration to reliable regions; Figs 9&10 demonstrates that adapting δ based on proxy uncertainty improves robustness.

**Essential References Not Discussed:**

Not that I'm aware of.

**Experimental Designs Or Analyses:**

Yes, esp. fig 2,3,5. The proxy model failure analysis confirms low correlation between proxy and oracle for OOD sequences, and the ablation studies on δ values confirm that conservative search outperforms unconstrained GFlowNet training.

**Methods And Evaluation Criteria:**

Yes.

**Other Comments Or Suggestions:**

N/A

**Other Strengths And Weaknesses:**

Strengths
- Intuitive adaptive δ mechanism balances novelty and conservativeness.
- Writing is clear and concise.
- Strong experimental validation across diverse sequence design tasks.

Weaknesses
- Does training on Dₜ₋₁ (containing more synthetic sequences) drift proxy model accuracy?
- Any intuition  on how extreme δ values (near 0 or 1) might affect performance? No major experiments needed, just some textual discussion would suffice.

**Questions For Authors:**

See above, Unclear if growing synthetic data proportion (from δ-CS) leads to accuracy drift in later rounds (potential model bias).

**Relation To Broader Scientific Literature:**

This paper has broad connection to offline-RL, active learning, and biological sequence design.

**Theoretical Claims:**

No theoretical guarantees on convergence or stability with increasing rounds, would be interesting to see some theoretical or experimental insights.

---

> ### Author Rebuttal · Authors · 2025-03-31
>
> > (Theoretical Claims) No theoretical guarantees on convergence or stability with increasing rounds, would be interesting to see some theoretical or experimental insights.
>
> Thanks for highlighting the importance of theoretical guarantees and analysis. We acknowledge that rigorous theoretical guarantees, such as formal convergence rates, are indeed valuable. However, deriving such guarantees for deep-learning-based active learning is exceptionally challenging—moreover, the proxy model and the generative policy interplay to explore the black-box landscape. Our primary goal is to provide a methodological and empirical contribution. We demonstrate the effectiveness of $\delta$-CS by presenting performance gains on DNA, RNA, and protein design tasks. These results clearly illustrate the practical benefits of our proposed approach across various domains. We will include this in our limitations of Section 7.
>
> ---
>
> > W1. Does training on $D_{t-1}$ (containing more synthetic sequences) drift proxy model accuracy?
>
> In the active learning setting, we assume that $D_{t-1}$ consists of annotated data with true oracle functions with limited accessiblity; e.g., we can query sequences with batch size of B = 128 at each round. Note that the predicted data $(x, f_{\phi}(x))$ is added to $D_{t-1}$ during the policy training.
>
> ---
>
> > W2. Any intuition on how extreme $\delta$ values (near 0 or 1) might affect performance? No major experiments needed, just some textual discussion would suffice.
>
> When $\delta$=0, no noise is injected into the offline data, so the generative policy is trained purely on the existing data without any additional exploration. In contrast, when $\delta$=1 the original offline data is completely replaced with noise, and the model fully relies on its own policy to denoise, meaning fully on-policy learning with no conservatism.
>
> ---
> > Q1. Unclear if growing synthetic data proportion (from $\delta$-CS) leads to accuracy drift in later rounds (potential model bias).
>
> At the start of each round, we reinitialize the generator model. During training, the model uses a proxy (acquisition function) to evaluate sequences, but we do not add them to our dataset. After training, we propose new sequences to be evaluated with oracle functions and then add those annotated results to the dataset, so the dataset always contains only real (annotated) data.
>
> ---
>
> **Thanks for your valuable comments**

---

### Official Review · Reviewer_6J1t · 2025-03-20

**Overall Recommendation:** 3

**Summary:**

This manuscript proposes a novel off-policy search strategy, δ-Conservative Search (δ-CS), that improves the reliability of GFlowNets for biological sequence design by controlling exploration according to proxy model confidence. The method randomly masks high-scoring offline sequences with probability δ, then relies on GFlowNets to denoise the masked tokens, keeping generated sequences closer to well-understood regions while still encouraging diversity. Experiments on DNA, RNA, and protein design tasks show that δ-CS consistently outperforms existing methods, including simpler off-policy baselines, by striking a better balance between exploration and exploitation when proxy models are uncertain.

**Claims And Evidence:**

I have no comments on this topic.

**Essential References Not Discussed:**

I have no comments on this topic.

**Experimental Designs Or Analyses:**

Please see the Part of  Strengths and Weaknesses.

**Methods And Evaluation Criteria:**

Please see the Part of  Strengths and Weaknesses.

**Other Comments Or Suggestions:**

I have no other comments.

**Other Strengths And Weaknesses:**

**Strengths**\
1.	The paper is generally well-written.

2.	The proposed method combines the diversity exploration of GFlowNets and the conservatism of evolutionary search. Through masking and adaptive delta mechanism, it effectively alleviates the problem of proxy model on out-of-distribution samples.

3.	Experimental evaluations across diverse biological sequence design tasks were performed to demonstrate both the generalizability and superior performance of the proposed method.


**Weakness**\
1.	The initial value and adjustment strategy of δ may depend on task experience (e.g., longer sequences require smaller δ), and there is a lack of general guidelines.

2.	The effectiveness of adaptive δ is highly dependent on the uncertainty estimation accuracy. If the estimation deviation is large, it may affect the performance.

3.	While δ-CS is designed for GFlowNets, its applicability to other types of generative models or RL algorithms in biological sequence design is not extensively explored.

**Questions For Authors:**

1.	Does the oracle f in the paper come from ground truth or a proxy evaluation model? If it comes from ground truth, how to evaluate the noised sequence that does not have ground truth?

**Relation To Broader Scientific Literature:**

This work will benefit protein design and drug design.

**Theoretical Claims:**

Please see the Part of  Strengths and Weaknesses.

---

> ### Author Rebuttal · Authors · 2025-03-31
>
> > W1. The initial value and adjustment strategy of $\delta$ may depend on task experience, and there is a lack of general guidelines.
>
> In this study, we simply set delta=0.5 for DNA/RNA (masking 4 to 7 tokens on average) and delta=0.05 for protein design (masking approximately 4 to 12 tokens). Our intention is to show that even with this simple setting without task-specific tunning, $\delta$-CS gives consistent improvements. Moreover, the results in Fig 12-14 in the appendix show that setting delta to mask approximately 1~12 tokens (i.e., 0.1 ≤ $\delta$ ≤ 0.5 in DNA/RNA, 0.01 ≤ $\delta$ ≤ 0.05 for protein design) is consistently beneficial compared to exploration without $\delta$-CS.
>
> As the reviewer mentioned, carefully setting delta based on experience and domain knowledge can bring even more improvements.
>
> ---
>
>  > W2. The effectiveness of adaptive $\delta$ is highly dependent on the uncertainty estimation accuracy. If the estimation deviation is large, it may affect the performance.
>
>
> We measure the uncertainty as the inconsistent proxy predictions using MC dropout or Ensemble. The key insight of the adaptive $\delta$ is to search the space more conservatively when the proxy gives inconsistent predictions on a certain datapoint compared to other datapoints. In addition, thanks to the scale parameter $\lambda$, we can adjust $\delta$ according to each point’s *relative* uncertainty. So, even if the estimation deviation (i.e., the discrepancy between predicted uncertainty and true uncertainty) is large, it does not overly degrade performance.
>
> ---
>
> > W3. While $\delta$-CS is designed for GFlowNets, its applicability to other types of generative models or RL algorithms in biological sequence design is not extensively explored.
>
>
> Thanks for the valuable feedback. As suggested, we conducted additional experiments using Soft Q-Learning (SQL) on the Hard TFBind-8 benchmark to demonstrate the broader applicability of our approach. As shown in the table below, $\delta$-CS significantly improves both exploration and performance in SQL, confirming that our method extends beyond the GFlowNet setting.
>
>
> | | t=1 | t=5 | t=10 |
> | - | - | - | - |
> | SQL | 0.531 (0.011) | 0.879 (0.014) | 0.936 (0.008) |
> | SQL + $\delta$-CS ($\delta$=0.5) |0.546 (0.013) | 0.975 (0.011) | 0.993 (0.007) |
>
>
> **The rationale for the experimental focus:** $\delta$-CS introduces a new off-policy exploration strategy under the unreliable proxy by conservatively leveraging known high-reward sequences. We initially focused on applying $\delta$-CS within GFlowNets because they offer a natural framework for structured sequence generation in biological domains. Moreover, there is a theoretical equivalence between GFlowNet training objectives (e.g., detailed balance, trajectory balance) and established off-policy max-entropy RL methods (e.g., Soft Q-Learning, Path Consistency Learning) when reward corrections are applied (Tiapkin et al., 2024; Deleu et al., 2024). This connection supports that the insights from our GFlowNet experiments can be extended to improvements in general off-policy RL settings.
>
>
>
> > Q1. Does the oracle $f$ in the paper come from ground truth or a proxy evaluation model? If it comes from ground truth, how to evaluate the noised sequence that does not have ground truth?
>
>
> In the case of TFBind-8, we use ground truth since all possible sequences are experimentally evaluated (Barrera et al., 2016) as the search space is relatively small ($4^8$). For other tasks, we assume a simulator or trained model with a larger dataset as our ground truth oracle, like various previous works (Sinai et al., 2020; Kirjner et al., 2023), and evaluate perturbed sequences using these oracle functions.
>
> Specifically, we use ViennaRNA (Lorenz et al., 2011) to evaluate the newly proposed RNA sequences. For GFP, the TAPE transformer model trained with 52,000, which is much larger than our initial dataset, is used as an oracle (Rao et al., 2019). In AAV, the oracle is built using comprehensive single-mutation data from AAV2 capsids, modeled additively (summing individual mutation effects with some noise), and applied to multiple target tissues across varying design lengths (Ogden et al., 2019).
>
> - Tiapkin et al. (2024) “Generative flow networks as entropy-regularized RL.”
> - Deleu et al. (2024) “Discrete probabilistic inference as control in multi-path environments.”
> - Sinai et al. (2020) “Adalead: A simple and robust adaptive greedy search algorithm for sequence design.”
> - Kirjner et al. (2023) “Improving protein optimization with smoothed fitness landscapes.”
> - Barrera et al. (2016) “Survey of variation in human transcription factors reveals prevalent DNA binding changes.”
> - Lorenz et al. (2011) “ViennaRNA Package 2.0.”
> - Rao et al. (2019) “Evaluating protein transfer learning with TAPE.”
> - Ogden et al. (2019) “Comprehensive AAV capsid fitness landscape reveals a viral gene and enables machine-guided design.”
>
> **Thanks for your valuable comments**

---

### Decision · Program_Chairs · 2025-05-01

**Decision:**

Accept (poster)

**Comment:**

This manuscript proposes a conservative search strategy for GFlowNets in the context of biological sequence design. The approach constrains the number of mutations during exploration based on sequence length and model uncertainty, effectively focusing the search around the initial sequence. Experimental results demonstrate that this strategy improves the stability of GFlowNet training and leads to better performance compared to GFlowNet-AL and traditional baselines. The results suggest that simple, principled restrictions on the search space can be highly beneficial, especially in domains where functionality tends to concentrate near a reference sequence.

The reviewers agree that the paper is clearly written, with well-motivated methodology and thorough experimental setup. The authors provided thoughtful clarifications and new experiments during the rebuttal, which addressed most concerns raised in the initial reviews.

I recommend acceptance and encourage the authors to incorporate the additional experimental results and clarify the rationale behind their experimental choices in the final version.